# A Recovery Guarantee for Sparse Neural Networks

**Sara Fridovich-Keil** *
School of Electrical and Computer Engineering
Georgia Institute of Technology
sfk@gatech.edu

**Mert Pilanci**
Department of Electrical Engineering
Stanford University
pilanci@stanford.edu

## Abstract

We prove the first guarantees of sparse recovery for ReLU neural networks, where the sparse network weights constitute the signal to be recovered. Specifically, we study structural properties of the sparse network weights for two-layer, scalar-output networks under which a simple iterative hard thresholding algorithm recovers these weights exactly, using memory that grows linearly in the number of nonzero weights. We validate this theoretical result with simple experiments on recovery of sparse planted MLPs, MNIST classification, and implicit neural representations. Experimentally, we find performance that is competitive with, and often exceeds, a high-performing but memory-inefficient baseline based on iterative magnitude pruning. Code is available at `https://github.com/voilalab/MLP-IHT`.

## 1 Introduction

Consider the task of training a sparse multilayer perceptron (MLP). We view this task through the lens of sparse signal recovery, in which the signal to be recovered is the vectorized MLP weights, most of which are zero — so exact recovery requires finding the indices and values of the few nonzero MLP weights. Are these weights uniquely identifiable from training data? Can they be recovered efficiently in both memory and iteration complexity? For scalar-output, two-layer MLPs we answer both questions in the affirmative, proving what is to our knowledge the first recovery guarantee for sparse MLP weights.

Large neural networks are widely used as universal function approximators (Hornik et al., 1989), but as model size grows networks require ever larger memory and compute time to train (Kaplan et al., 2020). Although large networks tend to be trainable to the highest quality, trained network weights are often highly compressible, e.g. by pruning, allowing for dramatic savings in memory and computation at inference time (Cheng et al., 2024). While sparse and high-performing networks are known to exist, efficiently optimizing them is an open challenge. Existing approaches often compromise either memory efficiency—requiring memory to first train a dense network (Frankle & Carbin, 2019; Saikumar & Varghese, 2024; Gharatappeh & Sekeh, 2025)—or quality, failing to match the performance of dense counterparts (Frankle et al., 2021; Saikumar & Varghese, 2024). While some strategies can empirically balance efficiency and quality (Parger et al., 2022; Jin et al., 2016; Damadi et al., 2024), all existing approaches to sparse network training are heuristic in nature and lack formal guarantees of weight recovery.

At the same time, the compressed sensing literature is rich with theoretically-justified algorithms to leverage sparsity in large-scale optimization tasks (see e.g., Wright & Ma (2022) for an accessible overview). However, these results are typically designed for linear models and convex optimization, and do not directly apply to recovery of sparse MLP weights (Tropp, 2004; Khanna & Kyrillidis, 2018; Aghazadeh et al., 2018).

Our work bridges this gap by leveraging the recent development of a convex reformulation of MLPs (Pilanci & Ergen, 2020a; Ergen & Pilanci, 2024), which allows us to apply strong results from sparse signal estimation (Jain et al., 2014) to the task of training a sparse MLP. In its convex reformulation,

---

*This work was initiated as a postdoc at Stanford, and completed as an assistant professor at Georgia Tech.

sparse MLP optimization can be viewed as a highly structured linear sensing problem in which the network weights are the signal to be recovered. We show that, when the training data consists of network evaluations at random Gaussian sample points, this highly structured sensing matrix satisfies (with high probability) the classic restricted strong convexity and restricted smoothness conditions that suffice to enable efficient sparse recovery via a simple projected gradient descent method known as Iterative Hard Thresholding (IHT). Concretely, we make the following contributions:

- We prove the first sparse recovery result applicable to ReLU MLPs, focusing on the case of a shallow scalar-output network and random Gaussian data. Our result includes both unique identifiability of sparse network weights as well as a high-probability guarantee of efficient recovery of these weights via IHT, building on a result from Jain et al. (2014).

- We demonstrate in a suite of illustrative small-scale experiments that IHT indeed tends to outperform a strong but memory-inefficient baseline of iterative magnitude pruning (IMP) (Frankle & Carbin, 2019), recovering higher-performing sparse networks while using less memory during optimization. Our experiments include both 2-layer and 3-layer MLPs with both scalar and vector valued outputs, extending beyond the regime of our theoretical results.

## 2 RELATED WORK

### 2.1 SPARSE NEURAL NETWORKS

Prior work has shown that, in diverse contexts, a large neural network may be well approximated by a sparse subnetwork, for example with only 10% of the original parameters left nonzero (Frankle & Carbin, 2019; Nowak et al., 2023). Sparse networks are far cheaper and faster to evaluate and store, making them attractive for applications on edge and resource-constrained platforms as well as for democratizing access to large foundation models. Moreover, in many cases a sparse subnetwork can even outperform the prediction accuracy (Frankle & Carbin, 2019) and out-of-distribution robustness (Diffenderfer et al., 2021; Wu et al., 2024) of the original dense network.

However, sparse networks are notoriously difficult to optimize. Existing approaches to finding sparse networks fall into three categories: iterative pruning (Frankle & Carbin, 2019; Liu et al., 2024), pruning at initialization (Wang et al., 2022; Frankle et al., 2021), and dynamic sparse training (Jin et al., 2016; Ji et al., 2024; Nowak et al., 2023; Damadi et al., 2024; Kusupati et al., 2020). Respectively, these approaches tend to be high-performing but require high memory during optimization, memory and computation efficient to optimize but with reduced final model performance, and efficient but heuristic to optimize to reasonable final performance. None of the existing sparse network optimization paradigms come with theoretical understanding or recovery guarantees.

Prior theoretical results for sparse neural networks are present in Boursier & Flammarion (2023) and Ergen & Pilanci (2021) (see Lemma 10 therein), which derive conditions under which the sparsest two-layer MLP may be recovered by minimizing the Euclidean norm of the weights (i.e., applying weight decay). However, Boursier & Flammarion (2023) focuses on univariate data and Ergen & Pilanci (2021) considers sparsity of the second (output) layer weights, whereas our analysis considers arbitrary data dimension with a focus on sparsity of the first (hidden) layer weights. The recovery result in Ergen & Pilanci (2021) also requires fewer data points than dimensions, while our result does not. Further, the conditions in Ergen & Pilanci (2021) are based on the KKT optimality conditions of a semi-infinite convex formulation (Hettich & Kortanek, 1993) and are not straightforward to verify, nor is a tractable recovery algorithm presented in Ergen & Pilanci (2021). In contrast, our guarantee of sparse weight recovery relies on verifiable and satisfiable conditions that we show hold with high probability under random training data, and we prove that an iterative algorithm (iterative hard thresholding) achieves successful recovery of sparse neuron weights.

### 2.2 CONVEX NEURAL NETWORKS

Recent work has revealed an equivalence between training shallow (Pilanci & Ergen, 2020a) or deep (Ergen & Pilanci, 2024) neural networks and solving convex optimization problems defined by network architectures. The core idea involves enumerating or sampling neuron activation paths to form a fixed dictionary, whose coefficients are optimized via convex programming.

Specifically, a two-layer ReLU network approximates labels $y$ using the nonconvex form $y \approx \sum_{j=1}^{p}(Xu_j)_+ v_j$, where $U = [u_1, ..., u_p]$ and $v$ are the network weights and $X$ is the data matrix. Instead, the convex formulation uses activation patterns $D_i = \text{Diag}(\mathbb{I}[Xu \geq 0])$ enumerated over all $u$ to express the same network as

$$y \approx \sum_{i=1}^{P} D_i X(\tilde{w}_i - w_i), \tag{1}$$

subject to $(2D_i - I_n)X\tilde{w}_i \geq 0$ and $(2D_i - I_n)Xw_i \geq 0$ for all $i$. Optimal values of the nonconvex weights $U$ and $v$ can be recovered from optimal values of the convex optimization parameters $\tilde{w}$ and $w$. Note that we use the term *activation pattern* to refer to a binary pattern whose length matches the number of training examples, and whose values denote which training examples are attended to by a particular neuron (each neuron has its own activation pattern). The total number of activation patterns $P$ derived from all possible $u$ is bounded exponentially in the data rank $r$, typically requiring subsampling for computational tractability. However, assuming sparsity in weights dramatically reduces the number of possible patterns, enabling exact convex optimization for large-scale datasets. Section 3 describes how we adapt and specialize this convex MLP reformulation for sparse networks in our theory and experiments.

## 2.3 ITERATIVE HARD THRESHOLDING (IHT)

Iterative Hard Thresholding (IHT) is a special case of projected gradient descent, in which the projection is onto the nonconvex set of sparse vectors. For large-scale sparse recovery problems, IHT and additive algorithms such as basis pursuit and matching pursuit (Tropp, 2004) are often the only feasible algorithms, due to their memory efficiency compared to convex relaxations such as LASSO. IHT is also well-studied theoretically and comes with convergence guarantees both in its classic implementation (Blumensath & Davies, 2009; 2010; Jain et al., 2014) and accelerated variants (Blumensath, 2012; Khanna & Kyrillidis, 2018). Some results also exist for a variant of IHT augmented with a count sketch data structure (Aghazadeh et al., 2018), which can expand the regimes of sparsity under which IHT enjoys successful recovery. Of these theoretical results for sparse recovery by IHT, most require the measurement matrix to satisfy either the restricted isometry property (RIP) with a small enough RIP constant, or restricted strong convexity and restricted smoothness properties with a small enough condition number; these conditions are too strict for the sparse MLP weight recovery task we consider.

However, Jain et al. (2014) proved a more general sparse recovery result for IHT, showing recovery under restricted strong convexity and restricted smoothness with an arbitrary finite condition number. Jain et al. (2014)'s result holds for classic IHT with the relaxation that the hard thresholding step of IHT must project onto a larger sparsity level than that of the true signal, where the inflation factor grows with condition number. Our theoretical results build on this result to show that the task of recovering sparse MLP weights can be reformulated so as to satisfy the restricted strong convexity and restricted smoothness properties in expectation over Gaussian data, allowing us to show that IHT is guaranteed to recover the weights of a planted sparse MLP.

## 3 PRELIMINARIES

Consider a ReLU neural network with vector-valued input, scalar output, and a single hidden layer. We use $X \in \mathbb{R}^{n \times d}$ to denote the (Gaussian) data matrix with $n$ data points and data (input) dimension $d$. We denote the ground truth labels or values as $y \in \mathbb{R}^n$, and the neural network output as $\hat{y} \in \mathbb{R}^n$. The hidden weights of the 1-hidden-layer MLP are denoted $U \in \mathbb{R}^{d \times p}$ where $p$ is the width of the hidden layer. The columns of this weight matrix are $u_i \in \mathbb{R}^d, i = 1, \ldots, p$, and the second layer weights are $v_1, ..., v_p$.

We now describe convexifying the model by fusing the first and second layer weights of the nonconvex ReLU model $\sum_{j=1}^{p}(Xu_j)_+ v_j$. We can express this model as follows:

$$\hat{y} = \underbrace{[\text{diag}\,(\mathbb{I}\{Xu_1 \geq 0\})X \quad \ldots \quad \text{diag}\,(\mathbb{I}\{Xu_p \geq 0\})X]}_{A \in \mathbb{R}^{n \times dp}} \begin{bmatrix} u_1 v_1 \\ \vdots \\ u_p v_p \end{bmatrix} \tag{2}$$

where we use $\mathbb{I}\{x \geq 0\}$ to denote the elementwise indicator function, taking value $1$ at indices where $x_i \geq 0$ and value $0$ otherwise.

Training (e.g. with MSE loss) the 2-layer MLP in Equation (2) presents a nonconvex optimization problem, because the parameters $u_j$ appear in both the weight vector and the $A$ matrix. We convexify by simply replacing the $p$ weight vectors $u_i$ in the $A$ matrix with $p$ separate, fixed generator vectors $h_i \in \mathbb{R}^d$, and fusing the weights via $w_i = u_i v_i \,\forall i$. This parameterization was previously studied in Mishkin et al. (2022), where it was shown to yield the gated ReLU (GReLU) network class, which is equivalent in expressivity to standard ReLU networks. Here, we extend this approach and show that sparse ReLU networks can also be recovered using a similar strategy. We obtain

$$\hat{y} = \underbrace{[\mathrm{diag}\,(\mathbb{I}\{Xh_1 \geq 0\})X \quad \ldots \quad \mathrm{diag}\,(\mathbb{I}\{Xh_p \geq 0\})X]}_{A} \begin{bmatrix} w_1 \\ \vdots \\ w_p \end{bmatrix}; \tag{3}$$

in this formulation exact recovery amounts to finding the sparse vector $w^\star \in \mathbb{R}^p$ whose values are the weights of a ground truth, planted MLP. If we allow the effective hidden dimension $p$ to be very large (up to $2d\left(\frac{e(n-1)}{d}\right)^d$ (Pilanci & Ergen, 2020b)), we can choose a set of vectors $h_i$ such that $\{(\mathbb{I}\{Xh_i \geq 0\})\}_{i=1}^p$ is exactly the set of all possible distinct activation patterns achievable for dataset $X$. Recall that in our notation, the term *activation pattern* refers to a binary pattern whose length matches the number of training examples $n$, and whose values denote which training examples are attended to by a particular neuron (each neuron has its own activation pattern). Moreover, for sparse neural networks with at most $s'$ nonzero weights per hidden neuron, we have $p \leq 2s'\binom{d}{s'}\left(\frac{n}{s'}\right)^{s'}$ by a counting argument; this may be far fewer total activation patterns than needed to model dense weights. Consider a neuron whose weight vector has at most $s'$ nonzero entries. First, the support of this weight vector must be selected, which corresponds to choosing $s'$ input dimensions out of $d$, resulting in $\binom{d}{s'}$ possible choices. For each choice of these $s'$ dimensions, the neuron computes a linear threshold function in an $s'$ dimensional subspace of $\mathbb{R}^n$. A classical result in the theory of hyperplane arrangements (Stanley et al., 2007) shows that such a linear threshold function can generate at most $2\sum_{i=0}^{s'-1}\binom{n-1}{i} \leq 2\left(\frac{n}{s'}\right)^{s'}$ distinct activation patterns over $n$ data points. Multiplying the number of ways to select the support and the number of patterns per support, and incorporating a factor of $s'$ for indexing neurons, we arrive at the stated bound: $p \leq 2s'\binom{d}{s'}\left(\frac{n}{s'}\right)^{s'}$. With this large but fixed set of generator vectors $h_i$, we can solve a similarly large but fixed convex program to recover hidden weights $w_i$ corresponding to the globally optimal 2-layer nonconvex MLP.

Alternatively, we can operate with an arbitrary hidden dimension $m$ and select the generator vectors $h_i$ at random such that the activation patterns $\{(\mathbb{I}\{Xh_i \geq 0\})\}_{i=1}^m$ are a random subset (drawn without replacement) of all $p$ possible activation patterns. In our theoretical results (Section 4) we assume patterns are enumerated; in our experiments (Section 5) we sample $m \leq p$ patterns using random generator vectors.

## 4 THEORETICAL RESULTS

Consider the sparse recovery problem defined by Equation (3) of the form $y = Aw^\star$ for some unknown vector $w^\star$, with sensing matrix

$$A := [\mathrm{diag}\,(\mathbb{I}\{Xh_1 \geq 0\})X \quad \ldots \quad \mathrm{diag}\,(\mathbb{I}\{Xh_p \geq 0\})X] \in \mathbb{R}^{n \times dp}.$$

Our main result leverages connections between sparse recovery methods and convex formulations of ReLU networks. For simplicity, we will assume that the data matrix $X \in \mathbb{R}^{n \times d}$ has entries drawn i.i.d. $\mathcal{N}(0,1)$; a similar effect may be achieved in practice by data whitening. We also assume that the columns of $A$ are unit-normalized before optimization.

To recover the planted weights $w^\star$, we consider the following simple variant of the classic Iterative Hard Thresholding (IHT) algorithm,

$$w^{k+1} = H_{\tilde{s}}(w^k - \eta A^T(Aw^k - y)). \tag{4}$$

Here $\eta > 0$ is a step size parameter and the hard thresholding operation $H_{\tilde{s}}$ is a projection onto the set of $\tilde{s}$-sparse vectors, where $\tilde{s} > s$ following Jain et al. (2014). In Lemma 1 we show that $A$

satisfies restricted strong convexity and restricted smoothness with high probability over the random data $X$, making the sparse MLP weights uniquely identifiable. In Theorem 1 we show that IHT efficiently recovers these sparse MLP weights.

Suppose that $y = \sum_{i=1}^p (Xu_i^\star)_+ v_i^\star = Aw^\star$ is the planted neural network model. Recall that the relation between the standard and fused form of the weights is $w^\star = [u_1^\star v_1^\star, ..., u_p^\star v_p^\star]$ where $\text{sign}(Xh_i) = \text{sign}(Xu_i^\star) \forall i$ as defined in (2) and (3). Assumption 1 gives conditions on a planted network under which we can ensure exact recovery of its weights.

**Assumption 1** (Properties of the planted sparse network). *Assume that either*

*(a) $u_i^\star \in \{-1, 0, 1\}^d$, $\|u_i^\star\|_0 = k$, $v_i^\star \in \mathbb{R} \, \forall i \in [p]$ and $kp \leq s$, or*

*(b) $u_i^\star \in \mathbb{R}^d$, $\|u_i^\star\|_0 = s_i \in [s_{min}, k]$, $v_i^\star \in \{-1, 1\} \, \forall i \in [p]$ and $\sum_{i=1}^p s_i \leq s$ holds.*

Both parts of Assumption 1 have to do with what values the planted MLP weights can take, and both parts restrict the number of nonzero hidden weights. Assumption 1(a) requires that the nonzero hidden weights take binary values, but allows the output layer weights to take any real values. Assumption 1(b) captures the more relaxed and common scenario in which the nonzero hidden weights can take any real values, but the output layer weights are restricted to $\pm 1$, since the flexibility to model any real value is already captured by the hidden layer weights.

In Appendix E we show that Assumption 2 follows from either of Assumption 1(a) and 1(b) with high probability, and we give weight constructions that satisfy each option in Assumption 1. We note that only Assumption 2 is used in our proof of convergence and sparse recovery; Assumption 1 is sufficient for Assumption 2 but may not be necessary. Likewise, we show that Assumption 2 is sufficient for sparse recovery but we do not prove that it is necessary.

**Assumption 2** (Properties of activation patterns). *Let $D_i = \text{diag}(\mathbb{I}\{Xh_i \geq 0\}) \in \mathbb{R}^{n \times n}$, with $\{D_i\}_{i=1}^p$ as the set of all such distinct activation patterns possible with data $X \in \mathbb{R}^{n \times d}$, whose entries are drawn i.i.d. $\sim \mathcal{N}(0, 1)$. We assume the following properties about this set of enumerated activation patterns:*

*1. $\text{Tr} \, D_i \geq \varepsilon n$ for all $i \in [p]$, for some $\varepsilon \in (0, 1)$.*

*2. For all $i \neq i'$, the diagonals of $D_i$ and $D_{i'}$ differ in at least $\gamma n$ positions, for some $\gamma \in (0, 1)$.*

*In the appendix we prove that both of these hold with high probability under Assumption 1. Specifically, Assumption 2.1 holds with probability at least $1 - pe^{-n\left(\frac{1-\varepsilon}{128} - \mathcal{H}(\varepsilon)\right)}$, as long as $n \geq 4k$. Here $\mathcal{H}$ denotes binary entropy. Assumption 2.2 follows from Assumption 1(a) with probability at least $1 - 2e^{-c\delta^2 n}$, as long as $n \geq C\delta^{-6} w(K)^2$ and $k \leq \frac{0.69}{\pi(\gamma + \delta)}$. Here $c$ and $C$ are positive absolute constants, $\delta > 0$, and $w(K)$ is the normalized Gaussian mean width of a subset $K \subseteq \mathbb{R}^d$, where $K$ represents the set of (normalized) neuron weights that satisfy Assumption 1(a). Assumption 2.2 likewise follows from Assumption 1(b) with probability at least $1 - 2e^{-c\delta^2 n} - \tilde{\epsilon}$ as long as $n \geq C\delta^{-6} w(K)^2$, where now $K$ represents the set of (normalized) neuron weights that satisfy Assumption 1(b). Note that $\tilde{\epsilon}$ and some additional restrictions on $s_{\min}$ and $k$ are described in the appendix proof.*

**Remark 1** (Sample complexity). *Note that Assumption 2 requires the number of training examples $n \geq \max(4k, C\delta^{-6} w(K))$, where $k$ is the sparsity level of each neuron, $K$ is the set of (normalized) neuron weights that satisfy Assumption 1(a) or 1(b), $w(K)$ is its normalized Gaussian mean width, and $C$ is a positive absolute constant. This is a modest requirement that grows with the number of active (nonzero) neuron weights rather than the total number of neuron weights, enabling compressive sensing of sparse neuron weights.*

Below we show that Assumption 2 is sufficient to ensure recovery of sparse MLP weights, for a 2-layer scalar-output ReLU MLP. Intuitively, the first part of Assumption 2 requires that every neuron attends to at least an $\epsilon$ fraction of the training data, rather than fitting or overfitting to a tiny number of examples. Since our data covariates are assumed Gaussian, this first part of Assumption 2 enables a concentration argument. The second part of Assumption 2 requires that any two different neurons must attend to subsets of the training dataset that differ by at least a $\gamma$ fraction. Without

this requirement, neurons might be very similar to each other and thus more difficult to distinguish and recover correctly during optimization. This second portion of Assumption 2 bears similarity in spirit with the incoherence property common in compressive sensing.

**Lemma 1** (Restricted strong convexity and restricted smoothness). *Let $A \in \mathbb{R}^{n \times dp}$ be as defined in Equation* (3)*, with the modification that all columns are normalized to have unit $\ell_2$ norm. Assume that entries of the data matrix $X \in \mathbb{R}^{n \times d}$ are drawn i.i.d. $\mathcal{N}(0,1)$ and Assumption 2 holds. Consider an index set $S \subseteq [dp]$ with $|S| = s \leq n$, and the induced $s \times s$ matrix $A_S^T A_S$. For any $\delta \in (0, \frac{\varepsilon}{1-\gamma})$, with probability at least $1 - 2s(s-1)\exp\left(\frac{-c\delta^2 \varepsilon \sqrt{n}}{1+\delta}\right) - 2s(s-1)\exp\left(\frac{-c\delta^2 n^{3/4}\varepsilon}{n^{1/4}+\delta}\right) - 8s(s-1)\exp\left(\frac{-c\delta^2 \varepsilon n}{1+\delta}\right)$,*

$$\alpha I_s \preceq A_S^T A_S \preceq \beta I_s$$

*with* $\alpha \geq 1 - \dfrac{1+\delta}{1-\delta}\sqrt{1-\gamma} - \dfrac{s}{n^{1/4}(1-\delta)}; \quad \beta \leq 1 + \dfrac{1+\delta}{1-\delta}(s-1)\sqrt{1-\gamma} + \dfrac{s}{n^{1/4}(1-\delta)}.$

*Here $\varepsilon$ and $\gamma$ are the same as in Assumption 2, and $c$ is a positive universal constant.*

Lemma 1 ensures that the condition number of $A$, restricted to any set of $s \leq n$ columns, is finite and bounded above by $\sqrt{\beta/\alpha}$. The condition number shrinks as $\gamma$ grows, because this enforces greater separation (incoherence) between columns of $A$. The conditioning worsens with increasing $s$, as this increases the number of columns in $A_S$ and thus the potential for a coherent pair of columns. Theorem 1 ensures that IHT recovers planted sparse weights regardless of this condition number (as long as it is finite), though the rate of convergence slows with increasing condition number.

**Theorem 1** (IHT recovers sparse MLP weights). *Suppose that Assumption 2 holds, the data matrix $X \in \mathbb{R}^{n \times d}$ has entries drawn i.i.d. $\mathcal{N}(0,1)$, the activation patterns $D_i = \mathrm{diag}\left(\mathbb{I}\{Xh_i \geq 0\}\right)$ in the sensing matrix $A$ are enumerated to include all unique patterns that can result from $\|h\|_0 \leq s$, the columns of $A$ are pre-normalized in $\ell_2$ norm, and the planted neural network weights satisfy $\|w^\star\|_0 \leq s$. Consider the following variant of Iterative Hard Thresholding (IHT) to minimize the MSE objective $f(w^k) = \frac{1}{2}\left\|Aw^k - y\right\|_2^2$:*

$$w^{k+1} = H_{\tilde{s}}\left(w^k - \eta A^T(Aw^k - y)\right), \tag{5}$$

*where $\tilde{s} \geq 32\left(\frac{\beta}{\alpha}\right)^2 s$, $\eta = \frac{2}{3\beta}$, and $\alpha, \beta$ are the restricted strong convexity and restricted smoothness constants from Lemma 1 corresponding to sparsity level $2\tilde{s}+s$. With the same high probability as in Lemma 1, after $K = \mathcal{O}\left(\frac{\beta}{\alpha}\log\left(\frac{f(w^0)}{\epsilon}\right)\right)$ steps, IHT finds sparse weights $w^K$ such that*

$$f(w^K) - f(w^\star) \leq \epsilon \quad \text{and} \quad \|w^K - w^\star\|_2^2 \leq 2\alpha^{-1}\epsilon.$$

**Remark 2.** *Theorem 1 is, to our knowledge, the first sparse recovery result that applies to sparse neural network weights. It extends Lemma 1 to show that sparse MLP weights are not only uniquely identifiable with high probability from a network's behavior on random data, but that these sparse weights may be recovered efficiently by IHT with high probability. If the underlying function mapping data points to values is indeed a planted sparse MLP, recovery of the weights of this sparse MLP also guarantees generalization, in the sense that the labels of fresh data following the same function will be perfectly predicted by the sparse MLP recovered by IHT.*

Proofs of all theoretical results may be found in the appendix.

## 5 EXPERIMENTAL RESULTS

Our experiments compare the performance of IHT and a strong MLP-pruning baseline method, iterative magnitude pruning (IMP), the algorithm from the Lottery Ticket Hypothesis (Frankle & Carbin, 2019), at training sparse MLPs. While these experiments are intended to complement and validate our sparse recovery theoretical results, they also extend beyond the setting of Theorem 1 in several respects, to demonstrate that IHT empirically recovers high-performing sparse MLPs even under more flexible settings than those for which we can prove sparse recovery succeeds.

Specifically, our range of experiments for IHT include (i) both full-batch (deterministic) and mini-batch (stochastic) gradients, (ii) both scalar and vector-valued MLP outputs, (iii) both single-hidden-layer and deeper MLPs, (iv) both vanilla and accelerated IHT, (v) randomized (rather than enumerated) initialization for the sensing matrix $A$, for computational efficiency, and (vi) sequential convex

updates to $A$ during IHT, rather than keeping $A$ fixed as we do in our theoretical analysis. These sequential convex updates interpolate between the fully convex formulation in our theory and the nonconvex training that is standard practice for MLPs, enabling empirically strong performance for IHT even with a much smaller, randomly-initialized $A$ compared to what is required in Theorem 1. The extension of IHT to vector-output MLPs and deeper MLPs is enabled by employing a count-sketch datastructure (following Aghazadeh et al. (2018)) for noisy but memory-efficient estimation of all weights, to slightly relax the hard thresholding in vanilla IHT (which we do use for shallow, scalar-output MLPs closer to the setting of our theoretical guarantees). We also strengthen the IMP baseline by pruning only 10% of the weights in each iteration (rather than the default 20%), which allows IMP to spend extra time finding a higher-performing sparse network. The details of our experimental settings for both IHT and IMP are provided in Appendix A and in our open-sourced code. We also provide some ablations and variability experiments in Appendix B.

We present experimental results on three illustrative tasks: fitting a planted sparse MLP, classifying handwritten MNIST digits (Deng, 2012), and fitting an implicit neural representation to MNIST and CIFAR-10 images (Krizhevsky et al., 2009). In each task, we compare the performance of IHT (ours) and IMP (Frankle & Carbin, 2019), implemented as described in Appendix A. For all figures, we show heatmaps comparing model performance as a function of the hidden dimension $m$ (vertical axis) and sparsity level $s$ (horizontal axis). We emphasize that IMP requires first training a dense MLP and then iteratively pruning it to achieve sparse weights, whereas IHT optimizes sparse weights directly and thus has far smaller memory requirements during training.

Results on scalar-output and vector-output planted sparse MLPs are presented in Figure 1 and Figure 2, respectively. Within Figure 1 and Figure 2, the left two subfigures compare IHT and IMP on 2-layer (1-hidden-layer) sparse MLPs while the right two subfigures compare IHT and IMP on 3-layer (2-hidden-layer) sparse MLPs. For the planted MLP fitting tasks, we optimize a sparse MLP with hidden dimension $m$ (heatmap vertical axis) and a budget of $s$ nonzero weights (heatmap horizontal axis) to match the input-output behavior of an unknown planted model of the same architecture and sparsity. Specifically, we draw random sparse weights and use these to generate a dataset ($X \in \mathbb{R}^{50000 \times 100}, y \in \mathbb{R}^{50000,c}$), for $c \in \{1, 10\}$. For planted MLP fitting tasks, we report peak signal to noise ratio (PSNR) at fitting this input-output behavior of the planted model. PSNR is defined as $\text{PSNR} = 10\log_{10}(I_{\max}^2/\text{MSE})$, where $I_{\max}$ is the largest magnitude value in the ground truth signal and MSE is the mean squared error; higher PSNR is better. If a certain setting of $m$ and $s$ yields a planted model whose outputs $y$ are all zero, we skip evaluation and report a PSNR of zero.

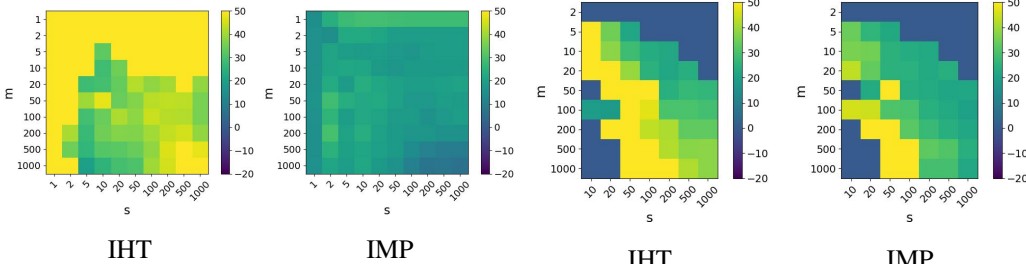

| IHT | IMP | IHT | IMP |

**Figure 1.** Average PSNR for fitting a planted one-hidden-layer (left) and two-hidden-layer (right) sparse scalar-output MLP of hidden dimension $m$ (vertical axis) and at most $s$ nonzero parameters (horizontal axis). Colorbar shows average PSNR over 3 random trials. IHT exhibits more robust performance than a strong but memory-inefficient iterative magnitude pruning (IMP) baseline (Frankle & Carbin, 2019).

Figure 3 presents results on MNIST digit classification. In Figure 3, all MLPs have one hidden layer; the left two subfigures compare IHT and IMP on binary classification and the right two subfigures compare IHT and IMP on 10-way classification. We consider both binary classification (digit 0 vs. 1) posed as a regression problem with MSE loss, as well as 10-way classification of all digits using cross-entropy loss and one-hot labels $y \in \mathbb{R}^{10}$. For MNIST classification, we report classification accuracy, the fraction of test digits correctly classified.

Although runtime is not a key component of our analysis or experiments, one trend worth noting is that the runtime for IHT is increasing in $s$, whereas the opposite is true for IMP. This is because IMP requires iterative retraining with gradual pruning, so more steps of retraining are required to reach

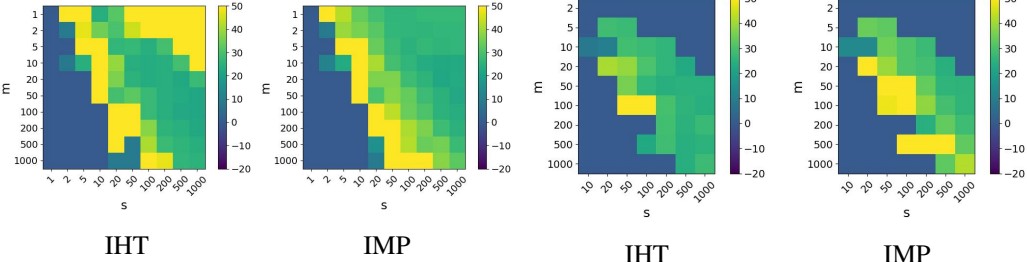

IHT            IMP                    IHT            IMP

**Figure 2.** Average PSNR for fitting a planted one-hidden-layer (left) and two-hidden-layer (right) sparse vector-output (10-dimensional output) MLP of hidden dimension $m$ (vertical axis) and at most $s$ nonzero parameters (horizontal axis). Colorbar shows average PSNR over 3 random trials. IHT is competitive with a strong but memory-inefficient iterative magnitude pruning (IMP) baseline (Frankle & Carbin, 2019).

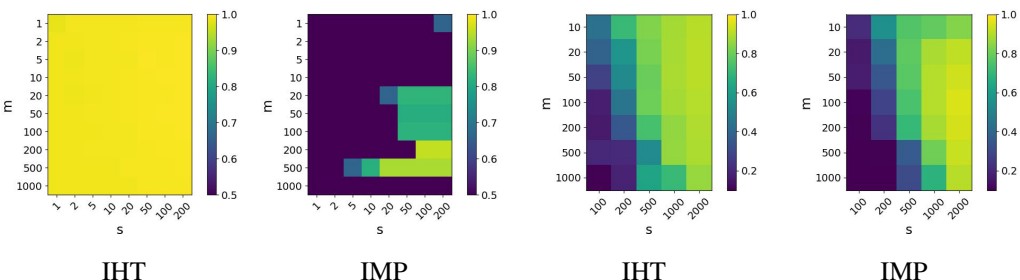

IHT            IMP                    IHT            IMP

**Figure 3.** Average binary (left) and 10-class (right) classification accuracy for handwritten MNIST digits with a 2-layer (one-hidden-layer) MLP of hidden dimension $m$ (vertical axis) and at most $s$ nonzero parameters (horizontal axis). Colorbar shows average classification accuracy over 3 random trials. IHT exhibits more robust performance than a strong but memory-inefficient iterative magnitude pruning (IMP) baseline (Frankle & Carbin, 2019).

a sparser network (with smaller $s$). In other words, IHT is fastest exactly where IMP is slowest. Runtime varies for both IHT and IMP as a function of problem parameters, so we provide a few illustrative examples, all evaluated on an NVIDIA A6000 GPU.

For binary MNIST classification using the smallest scalar-output model with $m = 1$ (hidden layer has a single neuron) and sparsity $s = 1$ (meaning that neuron can attend to a single pixel only), and 15 full-batch gradient steps, IHT reaches 98.85% test accuracy in 1.2 seconds, while IMP reaches 50% test accuracy (random chance) in 27.78 seconds. With $m = 10$ and $s = 100$, IHT reaches 99.2% accuracy in 1.66 seconds; IMP reaches 50.15% in 20.56 seconds. With $m = 100$ and $s = 1000$, IHT retains 99.2% accuracy in 8.4 seconds, and IMP achieves 77.66% accuracy in 20.91 seconds. For small, scalar-output MLPs IHT is dominant in memory, runtime, and accuracy.

However, for vector-output MLPs, deeper MLPs, and settings with large $s$, IHT (in its current implementation) can run more slowly than IMP, especially when using minibatch gradient updates. For full (10-class) MNIST classification, with 50 epochs, batch size 5000, $m = 10$, and $s = 1000$, IHT gets 88.73% test accuracy in 219.2 seconds, whereas IMP gets 77.66% accuracy in 68.98 seconds. For fitting a planted MLP with 10-dimensional output, with 15 epochs, full-batch gradients, $m = 10$, and $s = 10$, IHT reaches 48.67dB PSNR in 3.02 seconds while IMP reaches 24.81db PSNR in 28.46 seconds.

Figure 4 and Figure 5 present results on fitting MNIST and CIFAR-10 images, respectively, with an MLP-based implicit neural representation. Specifically, we use a fixed Fourier feature embedding with Gaussian-distributed frequencies followed by a ReLU MLP, following Tancik et al. (2020). We use the same Fourier features for embedding pixel coordinates for both IMP and IHT, and vary the optimization strategy for fitting the sparse MLP weights. Within Figure 4 and Figure 5, the left two subfigures compare IHT and IMP on 2-layer (1-hidden-layer) sparse MLPs while the right two subfigures compare IHT and IMP on 3-layer (2-hidden-layer) sparse MLPs.

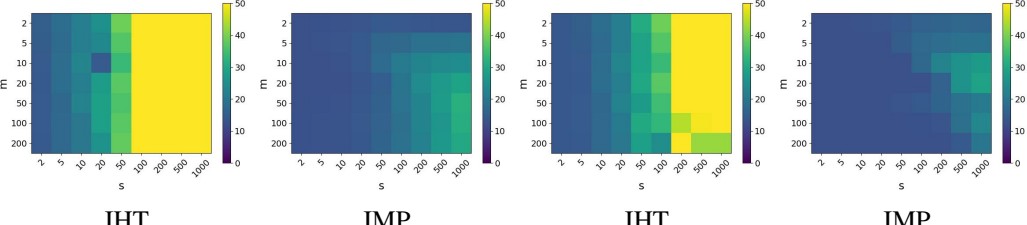

**Figure 4.** Average 1-hidden-layer (left) and 2-hidden-layer (right) PSNR for overfitting an MNIST digit image with an MLP-based implicit neural representation (Tancik et al., 2020) of hidden dimension $m$ (vertical axis) and at most $s$ nonzero parameters (horizontal axis). Colorbar shows average PSNR over 3 random trials. IHT exhibits more robust performance than a strong but memory-inefficient iterative magnitude pruning (IMP) baseline (Frankle & Carbin, 2019). We highlight that IHT exhibits stable recovery independent of $m$, in line with our theoretical results (see Remark 1). In contrast, IMP shows improved recovery with increasing $m$, likely because IMP here is solving a nonconvex optimization problem whose landscape is made more benign by increasing $m$.

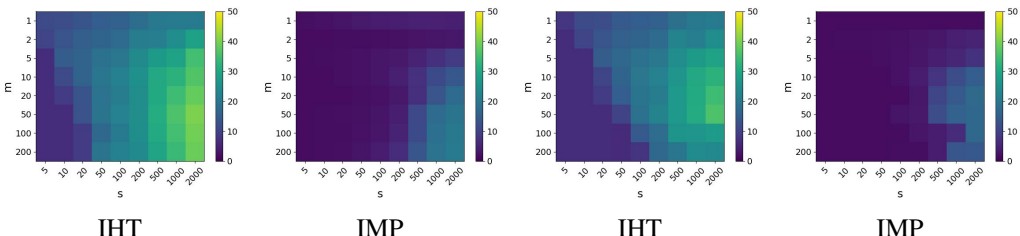

**Figure 5.** Average 1-hidden-layer (left) and 2-hidden-layer (right) PSNR for overfitting a CIFAR-10 digit image with an MLP-based implicit neural representation (Tancik et al., 2020) of hidden dimension $m$ (vertical axis) and at most $s$ nonzero parameters (horizontal axis). Colorbar shows average PSNR over 3 random trials. IHT exhibits more robust performance than a strong but memory-inefficient iterative magnitude pruning (IMP) baseline (Frankle & Carbin, 2019).

Across these experimental settings, we find that IHT almost always finds higher-performing sparse network weights compared to IMP, a strong baseline for pruning nonconvex MLPs (Frankle et al., 2021). Moreover, IHT uses a fixed parameter budget that scales with the sparsity level $s$ throughout optimization, whereas IMP requires initial training of a dense network whose parameter count grows with data dimension $d$ and hidden dimension $m$.

## 6 DISCUSSION

This work presents, to our knowledge, the first sparse recovery result applicable to the weights of a ReLU MLP. For nonnegative scalar-output MLPs, we show that sparse weights are uniquely identifiable and efficiently recoverable from measurements on random Gaussian data, with high probability. We complement this theoretical result with an empirical demonstration that a simple iterative hard thresholding algorithm can find sparser and higher-performing network weights compared to a strong network pruning baseline, while using far less memory during training.

**Limitations and future work.** Our results are subject to several limitations that we expect future work may address. Our theoretical results are restricted to shallow, scalar-output MLPs, and are shown to hold with high probability over Gaussian data rather than more general data distributions. As these are the first recovery results for sparse MLPs, we are optimistic that future work may extend our results to deeper, vector-output networks with more diverse architectures and data distributions. Our experiments also suggest that sequential convex optimization from random initialization can find high-performing sparse MLPs; extending our convex-formulation recovery result to sequential convex programming is of interest. Our IHT recovery result also inherits an inflated sparsity level $\tilde{s} > s$ from Jain et al. (2014); tightening this result is a compelling direction for further study. Finally, we encourage future work to refine and scale up our implementation of memory-efficient IHT training of sparse MLPs to enable both memory-efficient and fast training of high-performing sparse MLPs.

ACKNOWLEDGMENTS

This work was supported in part by the NSF Mathematical Sciences Postdoctoral Research Fellowship under award number 2303178, in part by the National Science Foundation (NSF) CAREER Award under Grant CCF-2236829, in part by the National Institutes of Health under Grant 1R01AG08950901A1, in part by the Office of Naval Research under Grant N00014-24-1-2164, and in part by the Defense Advanced Research Projects Agency under Grant HR00112490441. We are grateful to Gordon Wetzstein for helpful discussions on applications of sparse optimization.

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

APPENDICES

## A    EXPERIMENTAL METHODS

In this section we describe our experimental implementation of IHT as well as a network-pruning baseline algorithm, IMP (Frankle & Carbin, 2019). Our experiments are built on a mixture of Py-Torch and CuPy, and our code is available at `https://github.com/voilalab/MLP-IHT`.

**IHT updates.**    Our experiments use the classic IHT update rule $w^{k+1} = H_s(w^k - \eta_k \nabla f(w^k))$, where $f(w^k)$ is the objective function to be minimized. We do not inflate the projection sparsity level to the $\tilde{s}$ required in our theoretical analysis; doing so would likely further improve performance at the cost of inflating memory usage. For most of our experiments we use the mean squared error (MSE) objective with gradient $\nabla f_{MSE} = A^T(Aw^k - y)$. This yields the update rule in Equation (4), where in our experiments we use hard thresholding to enforce $s$-sparsity but not $r$-structure in the neurons. However, for our experiments on multiclass classification, we use the cross-entropy objective whose gradient is $\nabla f_{CE}(w^k) = A^T(\text{softmax}(Aw^k - y))$.

**Memory-efficient IHT implementation.**    A key strength of IHT is its memory efficiency, since only the nonzero weights and their indices need to be stored during optimization. However, achieving this memory efficiency requires careful implementation because each gradient $\nabla f(w^k)$ is a dense vector rather than a sparse one, and because the sensing matrix $A$ is huge. Our implementation is therefore blockwise. Instead of storing the entire matrix $A \in \mathbb{R}^{n \times dp}$ we generate each $n \times d$ block on the fly as it is needed. Instead of computing the entire gradient $\nabla f(w^k) \in \mathbb{R}^{dp}$ at once, we compute each $d$-dimensional block and apply it to the sparse iterate $w^{k+1}$ before computing the next block. Mathematically this is equivalent to computing the entire gradient and performing a single hard thresholding, but it can be far more memory efficient. The choice of block size is a design parameter that allows our IHT implementation strategy to trade off memory and computation time, allowing large sparse models to be trained under diverse hardware constraints.

**Sequential convex IHT.**    In our theoretical analysis, we assume that the $p$ activation patterns ($\mathbb{I}\{Xh_i \geq 0\}$) are enumerated to include all possible unique activation patterns based on fixed, sparse generator vectors $h_i$. This construction produces a large-scale but convex and very sparse optimization problem. In our experiments, for computational efficiency we instead solve a sequentially convex optimization that switches between the convex formulation in Equation (3) and the nonconvex formulation in Equation (2). We choose a fixed hidden dimension $m$ (rather than a larger enumerated dimension $p$) for the network weights, and frequently update the construction of the sensing matrix $A \in \mathbb{R}^{n \times dm}$ to maintain consistency with the weights $w \in \mathbb{R}^{dm}$ as they evolve during optimization, starting from a random initialization. In between these updates to $A$ the formulation is fixed and convex, hence the terminology of sequential convex optimization. We can equivalently view optimization in this sequential convex formulation as a time-varying dynamical system in which the sensing matrix $A$ is really $A_k$, as it depends on the current weight estimate $w^k$. We find the best performance arises from a two-stage optimization procedure in which we hold the generator vectors inside $A$ fixed at their random initialization until completion of the first epoch (pass through the training dataset), and then allow the generator vectors to update after each subsequent IHT iteration. Intuitively, this procedure stabilizes the first phase of optimization by maintaining convexity, and then allows for refinement of the sensing matrix once IHT has had the opportunity to enter a region of attraction around the global optimum.

**Vector-output MLPs.**    The formulation in Equation (2) and Equation (3) assumes a scalar-output MLP in which the output layer can be fused to the hidden layer weights. In an MLP with vector-valued output, we instead have $\hat{y} = (XW)_+\tilde{W}$, where as before $X \in \mathbb{R}^{n \times d}$ and $W \in \mathbb{R}^{d \times m}$ for hidden dimension $m$, but now $\hat{y} \in \mathbb{R}^{n \times c}$ and $\tilde{W} \in \mathbb{R}^{m \times c}$ for output dimension $c$. We can no longer fuse the output layer weights, so we optimize the following formulation:

$$\hat{y} = [\text{diag}\,(\mathbb{I}\{Xw_1 \geq 0\})X \quad \ldots \quad \text{diag}\,(\mathbb{I}\{Xw_m \geq 0\})X] \begin{bmatrix} w_1\tilde{w}_1^T \\ \vdots \\ w_m\tilde{w}_m^T \end{bmatrix} \tag{6}$$

where $w_i \in \mathbb{R}^d$ is a column of $W$ (as in the scalar-output case) and $\tilde{w}_i \in \mathbb{R}^c$ is a row of $\tilde{W}$. We use the chain rule to compute separate gradients for $W$ and $\tilde{W}$, computed blockwise and applied on-the-fly to update a global sparse weight representation for memory efficiency.

**Layerwise optimization for deeper MLPs.** Although it is possible to refine the formulation Equation (2) for deep MLPs (Ergen & Pilanci, 2024), for simplicity of implementation we optimize deep MLPs following a layerwise approach (Bengio et al., 2006; Karimi et al., 2024). For example, to optimize a 3-layer MLP (2 hidden layers plus an output layer), we proceed as follows. First, we optimize a 2-layer, vector-output MLP to find sparse weights $W \in \mathbb{R}^{d \times m}$ and $\hat{W} \in \mathbb{R}^{m \times c}$. We then discard $\hat{W}$ and freeze $W$, treating it as an input embedding while training a second 2-layer MLP, this time with input dimension $d = m$. We note that our results for IHT on deeper MLPs are slightly pessimistic, as our optimization procedure allocates some nonzero parameters to intermediate output layers $\tilde{W}$ that are not used in the final model, meaning that the final model performance is attained with strictly fewer active weights than the budgeted $s$. Nonetheless, IHT remains competitive despite this restriction (see Section 5).

**Count sketching.** For shallow scalar-output MLPs, we use the standard hard thresholding rule based on weight magnitudes, retaining the $s$ highest-magnitude entries in $W$ at each iteration. For deeper and vector-output MLPs, we follow Aghazadeh et al. (2018) and use an intermediate count sketch data structure to perform hard thresholding. Intuitively, we view the count sketch approach as a noisy but more memory efficient alternative to the deterministic sparsity inflation in our theoretical analysis. At every gradient step, we update the count sketch to maintain a noisy estimate of the full iterate, a vectorized concatenation of $W$ and $\tilde{W}$. We use a vector of dimension $4s \log(n/s)$ to represent the count sketch, balancing memory efficiency with the level of approximation error in the sketch. At each iteration of IHT, we find the $s$ entries in the count sketch with largest estimated magnitude, and store exact values for these entries. We observe an empirical tradeoff in the use of a count sketch: for shallow scalar-output networks where we have a single weight matrix $W$ to optimize, the approximation error introduced by the count sketch outweighs any benefit it brings by "softening" the hard thresholding operation. However, for vector-output networks or deeper networks where IHT must implicitly decide how to allocate a fixed parameter budget among $W$ and $\tilde{W}$, the count sketch allows IHT to make less myopic thresholding decisions that aggregate information from multiple gradient steps, the benefits of which appear to outweigh the cost of approximation error in the count sketch.

**Step size selection.** For our experiments with IHT, we use two different step size selection methods. For shallow scalar-output networks, we fuse the output layer weights following Equation (2). We can then compute an adaptive stepsize to minimize the mean squared error (MSE) objective function

$$\eta_k = \frac{\left\| A_{\text{supp}(w^k)}^T (y - Aw^k) \right\|_2^2}{\left\| A_{\text{supp}(w^k)} A^T (y - Aw^k) \right\|_2^2}, \tag{7}$$

following Blumensath & Davies (2010). However, as Equation (7) does not directly apply to vector-output networks, for these we use a fixed stepsize $\eta_k = \eta$ for both $W$ and $\tilde{W}$, and manually tune $\eta$. This manual tuning surely leaves room for improvement with an adaptive strategy, which we defer to future work.

**Accelerated IHT.** For shallow scalar-output networks, we use an accelerated IHT following Blumensath (2012), which defines an accelerated IHT as any algorithm that augments the classic IHT update with a refinement step to produce an iterate that is both sparse and has objective value no larger than that of the iterate produced by classic IHT. Specifically, after each IHT update we take a few gradient steps restricted to the current set of nonzero weights, to lower the objective value without changing the sparse support (Blumensath, 2012). We do not find an empirical benefit to this acceleration procedure for vector-output MLPs, so we perform acceleration only for IHT on scalar-output networks.

**IMP baseline.** We compare our IHT approach for optimizing sparse MLPs with iterative magnitude pruning (IMP) (Frankle & Carbin, 2019), a high-performing baseline method for pruning neural

networks that has been shown to find sparse networks that often match or exceed the quality of their dense counterparts. IMP begins by training a dense network, and then iteratively prunes (sets to zero) a constant fraction of the active (nonzero) weights based on magnitude, rewinds the remaining active weights to their initialization values, and retrains. IMP thus allows pruning a dense network to any desired sparsity level, but requires sufficient memory to train the dense network and sufficient computation time to iteratively retrain it during pruning. Although this IMP process is computationally costly, it provides a strong baseline of performance that can be achieved with a sparse network using existing methods. Frankle & Carbin (2019) suggest pruning 20% of the active weights at each iteration to balance computation and performance; we prune only 10% of the active weights at each iteration to maximize IMP performance and provide as strong a baseline as possible.

**Minibatches.**    For both IHT and IMP, each iteration operates on a minibatch of the full dataset. For IHT, we perform a minibatch update by subsampling the $n$ rows of both $A$ and $y$. For fitting a planted sparse MLP as well as for 10-way MNIST classification we use a minibatch size of 5000, and for binary MNIST classification we use a minibatch size of 1000. These settings all correspond to 10% of each training dataset per minibatch. For fitting an implicit neural representation to MNIST and CIFAR-10 images, we use full-batch updates as each image is small.

## B    META-EXPERIMENTS

Figures 6 to 10 parallel Figures 1 to 5 but include standard deviation over the three random trials; high-variance experiments tend to align with empirical phase transitions.

As our experiments use sequential convex updates to the matrix $A$ whereas our theoretical analysis assumes $A$ is static, we empirically evaluate the convergence of $A$ under our sequential convex updating strategy. Specifically, we consider the following setting: fitting a scalar-output planted sparse 2-layer MLP with input dimension 100, hidden dimension $m = 10$, and sparsity level $s = 500$, optimized over 100 steps of IHT with random initialization and sequential convex updates to $A$. We update the $A$ matrix every $k$ steps (in our main experiments $k = 1$), and report how the PSNR (fitting quality metric, higher is better) changes as a function of $k$. Note that when $k = 100$, $A$ is never updated because this experiment only runs for 100 steps. Results are reported in Table 1. We see that when $A$ is updated fairly frequently ($k < 5$) IHT converges and matches the planted model up to numerical precision. However, as $A$ is updated less frequently, the model no longer converges within 100 steps, and PSNR degrades with increasing $k$.

We also evaluate the convergence of $A$ more directly, by checking the first step at which the support of $A$ (which columns are active) stops changing; results are reported in Table 2. We find that as $k$ increases, $A$ takes longer to converge. As long as $k < 5$, $A$ still converges within 100 steps. However, when $k = 5$, $A$ does not converge within 100 steps, likely explaining why there is a dramatic drop in PSNR in Table 1 when $k$ increases from 4 to 5. If we instead continue optimization longer when $k = 5$, $A$ does converge at step 135, after which the PSNR increases to 161.44 (corresponding to machine precision for this task). From this set of experiments, we see that the frequency of sequential convex updates to A controls the convergence rate of IHT. This experiment also suggests a natural stopping criterion for IHT: when the support of A stops changing (though in our main experiments we use a fixed number of steps, for fair comparison to IMP).

## C    PROOF OF LEMMA 1

*Proof.* Our goal is to upper and lower bound the eigenvalues of $A_S^T A_S \in \mathbb{R}^{s \times s}$, where the full matrix $A \in \mathbb{R}^{n \times dp}$ is a column-normalized version of

$$[\text{diag}\,(\mathbb{I}\{Xh_1 \geq 0\})X \quad \ldots \quad \text{diag}\,(\mathbb{I}\{Xh_p \geq 0\})X] \in \mathbb{R}^{n \times dp},$$

$X \in \mathbb{R}^{n \times d}$ has i.i.d. $\mathcal{N}(0, 1)$ entries, and the index set $S \subseteq [dp]$ has $|S| = s \leq n$.

As singular values of $A_S$ are unaffected by column permutation, without loss of generality, we assume that columns of $A_S$ are ordered so that columns involving each $x_j$ are adjacent to each other. This ordering induces a block structure in $A_S^T A_S$; we refer to the $i, j$ block submatrix as $(A_S^T A_S)_{i,j}$. Using this block structure, we bound each type of entry in $A_S^T A_S$ with high probability

| k | PSNR |
|---|---|
| 1 | 161.44 |
| 2 | 161.44 |
| 3 | 161.44 |
| 4 | 161.44 |
| 5 | 36.63 |
| 6 | 31.85 |
| 7 | 31.84 |
| 8 | 28.49 |
| 9 | 29.88 |
| 10 | 28.38 |
| 15 | 27.23 |
| 20 | 25.85 |
| 25 | 24.75 |
| 30 | 24.74 |
| 35 | 23.32 |
| 40 | 23.35 |
| 50 | 22.02 |
| 60 | 21.95 |
| 70 | 21.96 |
| 80 | 21.97 |
| 90 | 21.89 |
| 98 | 21.57 |
| 100 | 19.88 |

**Table 1.** Fitting a scalar-output 2-layer MLP with input dimension 100, hidden dimension $m = 10$, and sparsity level $s = 500$, optimized over 100 steps of IHT with random initialization and sequential convex updates to $A$ every $k$ steps. In our main experiments $k = 1$; here we observe that fitting quality degrades with increasing $k$.

| k | convergence step |
|---|---|
| 1 | 43 |
| 2 | 64 |
| 3 | 75 |
| 4 | 84 |

**Table 2.** Fitting a scalar-output 2-layer MLP with input dimension 100, hidden dimension $m = 10$, and sparsity level $s = 500$, optimized over 100 steps of IHT with random initialization and sequential convex updates to $A$ every $k$ steps. In our main experiments $k = 1$; here we observe that $A$ converges more slowly with increasing $k$. We report "convergence step" as the first step of optimization at which the set of active columns of $A$ does not change, signaling support recovery.

and then use these entry-wise bounds to upper and lower bound the eigenvalues of $A_S^T A_S$ with high probability.

First, consider a block submatrix $(A_S^T A_S)_{i,i}$ on the diagonal of $A_S^T A_S$. Since the columns of $A$ are normalized, the diagonal entries of $(A_S^T A_S)_{i,i}$ are deterministically 1 for all $i$. The off-diagonal entries of $(A_S^T A_S)_{i,i}$ take the form $\frac{x_i^T D_j D_{j'} x_i}{\|D_j x_i\|_2 \|D_{j'} x_i\|_2}$. The numerator has $\mathbb{E}[x_i^T D_j D_{j'} x_i] = \operatorname{Tr} D_j D_{j'}$, while the denominator terms have expectation $\sqrt{\operatorname{Tr} D_j}$ and $\sqrt{\operatorname{Tr} D_{j'}}$, respectively. Applying Hanson-Wright (Theorem 2) to each quadratic form in this expression, we have

$$\frac{x_i^T D_j D_{j'} x_i}{\|D_j x_i\|_2 \|D_{j'} x_i\|_2} \leq \frac{\operatorname{Tr} D_j D_{j'} + t_1}{\sqrt{\operatorname{Tr} D_j - t_2} \sqrt{\operatorname{Tr} D_{j'} - t_3}} \tag{8}$$

with probability at least $1 - 2 \exp\left(\frac{-ct_1^2}{\operatorname{Tr} D_j D_{j'} + t_1}\right) - 2 \exp\left(\frac{-ct_2^2}{\operatorname{Tr} D_j + t_2}\right) - 2 \exp\left(\frac{-ct_3^2}{\operatorname{Tr} D_{j'} + t_3}\right)$, for a universal constant $c$.

Consider two regimes based on whether $\operatorname{Tr} D_j D_{j'}$ is less than or greater than $\varepsilon \sqrt{n}$. In the first regime, $\operatorname{Tr} D_j D_{j'} \leq \varepsilon \sqrt{n}$. We choose $t_1 = \varepsilon \sqrt{n}$, $t_2 = \delta \operatorname{Tr} D_j$, and $t_3 = \delta \operatorname{Tr} D_{j'}$ for $\delta \in (0, 1)$,

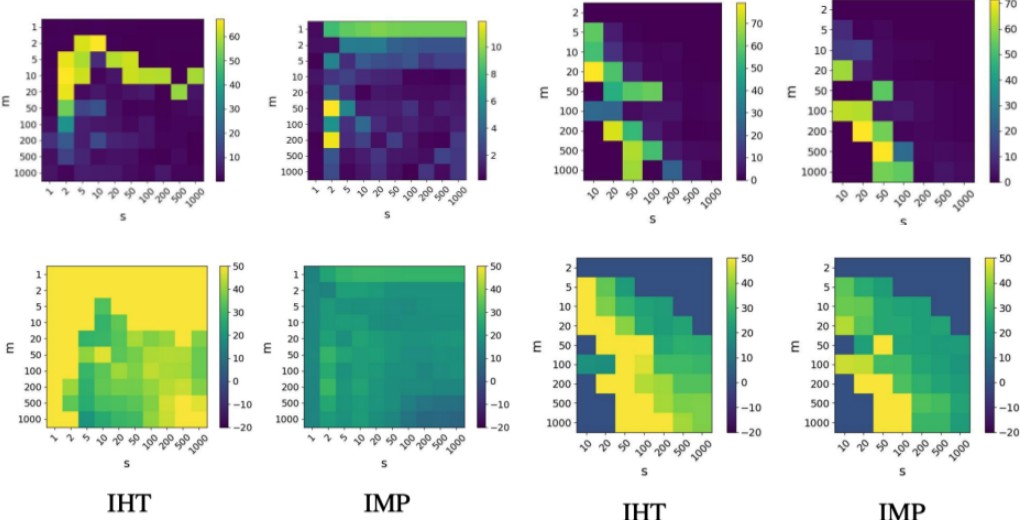

**Figure 6.** Standard deviation over the three random trials (top row) for each experiment reported in Figure 1 (bottom row).

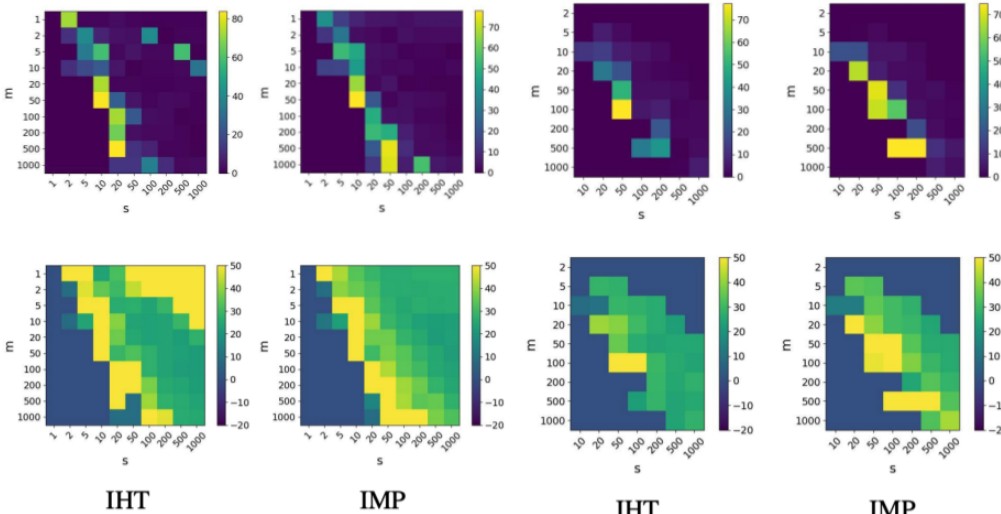

**Figure 7.** Standard deviation over the three random trials (top row) for each experiment reported in Figure 2 (bottom row).

which yields

$$\frac{x_i^T D_j D_{j'} x_i}{\|D_j x_i\|_2 \|D_{j'} x_i\|_2} \leq \frac{\operatorname{Tr} D_j D_{j'} + \varepsilon\sqrt{n}}{\sqrt{(1-\delta)\operatorname{Tr} D_j}\sqrt{(1-\delta)\operatorname{Tr} D_{j'}}}$$

with probability at least $1 - 2\exp\left(\frac{-c\varepsilon^2 n}{\operatorname{Tr} D_j D_{j'} + \varepsilon\sqrt{n}}\right) - 2\exp\left(\frac{-c\delta^2 \operatorname{Tr} D_j}{1+\delta}\right) - 2\exp\left(\frac{-c\delta^2 \operatorname{Tr} D_{j'}}{1+\delta}\right)$.

Since we are in the regime where $\operatorname{Tr} D_j D_{j'} \leq \varepsilon\sqrt{n}$ and, by Assumption 2, $D_j$ and $D_{j'}$ each have trace at least $\varepsilon n$, we have

$$\frac{x_i^T D_j D_{j'} x_i}{\|D_j x_i\|_2 \|D_{j'} x_i\|_2} \leq \frac{2}{(1-\delta)\sqrt{n}}$$

with probability at least $1 - 2\exp\left(\frac{-c\varepsilon\sqrt{n}}{2}\right) - 4\exp\left(\frac{-c\delta^2 \varepsilon n}{1+\delta}\right)$.

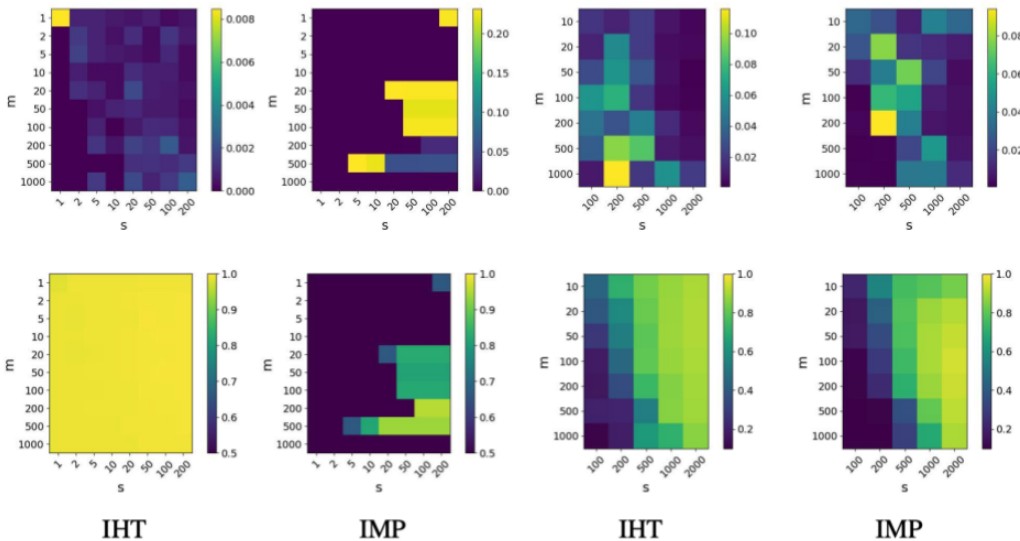

**Figure 8.** Standard deviation over the three random trials (top row) for each experiment reported in Figure 3 (bottom row).

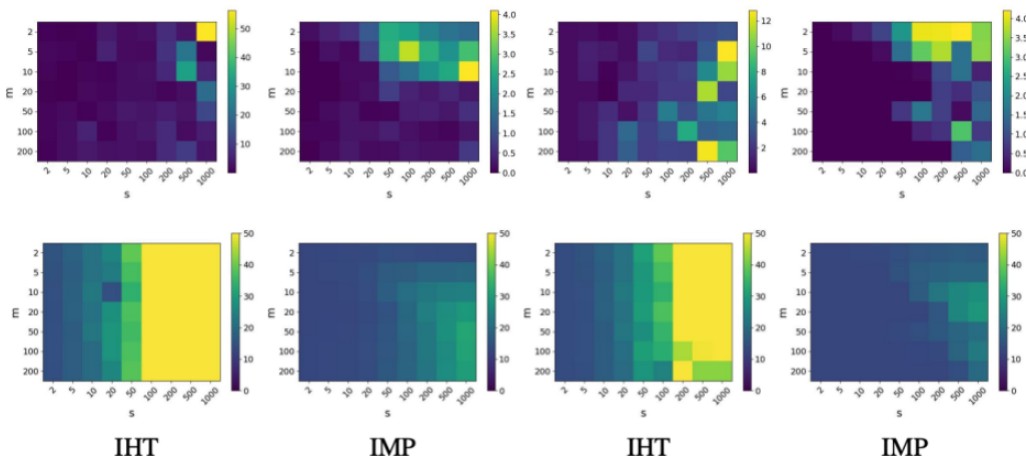

**Figure 9.** Standard deviation over the three random trials (top row) for each experiment reported in Figure 4 (bottom row).

In the second regime, $\text{Tr}\, D_j D_{j'} > \varepsilon \sqrt{n}$. In Equation (8) we choose $t_1 = \delta \, \text{Tr}\, D_j D_{j'}$, $t_2 = \delta \, \text{Tr}\, D_j$, and $t_3 = \delta \, \text{Tr}\, D_{j'}$, for $\delta \in (0,1)$, to yield

$$\frac{x_i^T D_j D_{j'} x_i}{\|D_j x_i\|_2 \|D_{j'} x_i\|_2} \leq \frac{(1+\delta)\, \text{Tr}\, D_j D_{j'}}{(1-\delta)\sqrt{\text{Tr}\, D_j}\sqrt{\text{Tr}\, D_{j'}}}$$

with probability at least $1 - 2\exp\left(\frac{-c\delta^2\, \text{Tr}\, D_j D_{j'}}{1+\delta}\right) - 2\exp\left(\frac{-c\delta^2\, \text{Tr}\, D_j}{1+\delta}\right) - 2\exp\left(\frac{-c\delta^2\, \text{Tr}\, D_{j'}}{1+\delta}\right)$. Without loss of generality, assume that $\text{Tr}\, D_j \leq Tr D_{j'}$. We are interested in upper bounding the quantity $\frac{x_i^T D_j D_{j'} x_i}{\|D_j x_i\|_2 \|D_{j'} x_i\|_2}$, which is maximized when all entries that take value 1 in $D_j$ also take value 1 in $D_{j'}$. This choice maximizes $\text{Tr}\, D_j D_{j'}$ for any fixed values of $\text{Tr}\, D_j$ and $\text{Tr}\, D_{j'}$. Let $\text{Tr}\, D_j = \xi n$. By Assumption 2, $D_j$ and $D_{j'}$ must take different values (one is 0 and the other is 1) in at least $\gamma n$ diagonal positions. Combining this with our observation of the maximizing arrangement of ones and zeros in $D_j$ and $D_j'$, for this arrangement we have that $\text{Tr}\, D_{j'} \geq \text{Tr}\, D_j + \gamma n = (\xi + \gamma)n$ and $\text{Tr}\, D_j D_{j'} = \text{Tr}\, D_j = \xi n$. Since we are in the regime where $\text{Tr}\, D_j D_{j'} > \varepsilon \sqrt{n}$ and, by Assumption

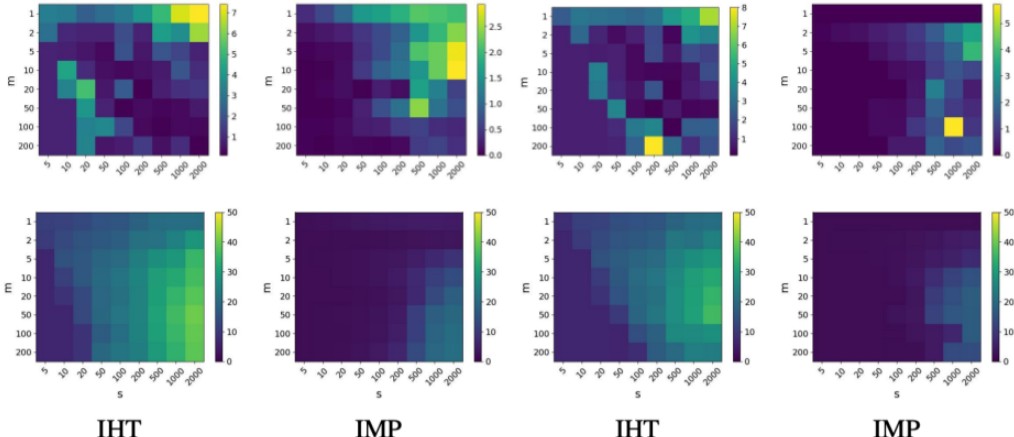

**Figure 10.** Standard deviation over the three random trials (top row) for each experiment reported in Figure 5 (bottom row).

2, $D_j$ and $D_{j'}$ each have trace at least $\varepsilon n$, we have

$$\frac{x_i^T D_j D_{j'} x_i}{\|D_j x_i\|_2 \|D_{j'} x_i\|_2} \leq \frac{(1+\delta)\sqrt{\xi}}{(1-\delta)\sqrt{\xi+\gamma}}$$

with probability at least $1 - 2\exp\left(\frac{-c\delta^2\varepsilon\sqrt{n}}{1+\delta}\right) - 4\exp\left(\frac{-c\delta^2\varepsilon n}{1+\delta}\right)$. This upper bound is increasing in $\xi$, which can take value at most $1-\gamma$ since $\operatorname{Tr} D_{j'} \geq (\xi+\gamma)n$ and by construction $D_{j'} \leq n$. We therefore set $\xi = 1 - \gamma$ to yield the bound:

$$\frac{x_i^T D_j D_{j'} x_i}{\|D_j x_i\|_2 \|D_{j'} x_i\|_2} \leq \frac{(1+\delta)\sqrt{1-\gamma}}{1-\delta},$$

which holds with probability at least $1 - 2\exp\left(\frac{-c\delta^2\varepsilon\sqrt{n}}{1+\delta}\right) - 4\exp\left(\frac{-c\delta^2\varepsilon n}{1+\delta}\right)$. Since this second-regime bound is independent of $n$ while the first-regime bound decays as $n^{-1/2}$, for large $n$ the second-regime bound dominates. For all $\delta \in (0,1)$ we also have that $1 - 2\exp\left(\frac{-c\delta^2\varepsilon\sqrt{n}}{1+\delta}\right) - 4\exp\left(\frac{-c\delta^2\varepsilon n}{1+\delta}\right) \leq 1 - 2\exp\left(\frac{-c\varepsilon\sqrt{n}}{2}\right) - 4\exp\left(\frac{-c\delta^2\varepsilon n}{1+\delta}\right)$. Therefore, we conclude that

$$0 \leq \frac{x_i^T D_j D_{j'} x_i}{\|D_j x_i\|_2 \|D_{j'} x_i\|_2} \leq \frac{(1+\delta)\sqrt{1-\gamma}}{1-\delta},$$

holds with probability at least $1 - 2\exp\left(\frac{-c\delta^2\varepsilon\sqrt{n}}{1+\delta}\right) - 4\exp\left(\frac{-c\delta^2\varepsilon n}{1+\delta}\right)$ for all off-diagonal entries of a diagonal block submatrix $(A_S^T A_S)_{i,i}$. Here we include a deterministic lower bound $\frac{x_i^T D_j D_{j'} x_i}{\|D_j x_i\|_2 \|D_{j'} x_i\|_2} \geq 0$, which holds because $D_j D_{j'}$ is a diagonal matrix with all entries nonnegative.

Next, we consider a block submatrix $(A_S^T A_S)_{i,i'}$ that is off the diagonal of $A_S^T A_S$. The entries of $(A_S^T A_S)_{i,i'}$ take the form $\frac{x_i^T D_j D_{j'} x_{i'}}{\|D_j x_i\|_2 \|D_{j'} x_{i'}\|_2}$, where $D_j$ and $D_{j'}$ may be the same or different. The numerator has $\mathbb{E}[x_i^T D_j D_{j'} x_{i'}] = 0$, while the denominator terms have expectation $\sqrt{\operatorname{Tr} D_j}$ and $\sqrt{\operatorname{Tr} D_{j'}}$, respectively. We bound this expression with high probability by applying Hanson-Wright to each term in the denominator, and asymmetric Hanson-Wright to the numerator:

$$\frac{|x_i^T D_j D_{j'} x_{i'}|}{\|D_j x_i\|_2 \|D_{j'} x_{i'}\|_2} \leq \frac{t_1}{\sqrt{\operatorname{Tr} D_j - t_2}\sqrt{\operatorname{Tr} D_{j'} - t_3}} \tag{9}$$

with probability at least $1 - 2\exp\left(\frac{-ct_1^2}{\operatorname{Tr} D_j D_{j'} + t_1}\right) - 2\exp\left(\frac{-ct_2^2}{\operatorname{Tr} D_j + t_2}\right) - 2\exp\left(\frac{-ct_3^2}{\operatorname{Tr} D_{j'} + t_3}\right)$, for a universal constant $c$. We choose $t_2 = \delta \operatorname{Tr} D_j$ and $t_3 = \delta \operatorname{Tr} D_{j'}$, for $\delta \in (0, 1)$, to yield

$$\frac{|x_i^T D_j D_{j'} x_{i'}|}{\|D_j x_i\|_2 \|D_{j'} x_{i'}\|_2} \leq \frac{t_1}{(1-\delta)\sqrt{\operatorname{Tr} D_j}\sqrt{\operatorname{Tr} D_{j'}}}$$

with probability at least $1 - 2\exp\left(\frac{-ct_1^2}{\operatorname{Tr} D_j D_{j'} + t_1}\right) - 2\exp\left(\frac{-c\delta^2 \operatorname{Tr} D_j}{1+\delta}\right) - 2\exp\left(\frac{-c\delta^2 \operatorname{Tr} D_{j'}}{1+\delta}\right)$. By Assumption 2, $\operatorname{Tr} D_j \geq \varepsilon n$ and $\operatorname{Tr} D_{j'} \geq \varepsilon n$, so we have

$$\frac{|x_i^T D_j D_{j'} x_{i'}|}{\|D_j x_i\|_2 \|D_{j'} x_{i'}\|_2} \leq \frac{t_1}{(1-\delta)\varepsilon n}$$

with probability at least $1 - 2\exp\left(\frac{-ct_1^2}{\operatorname{Tr} D_j D_{j'} + t_1}\right) - 4\exp\left(\frac{-c\delta^2 \varepsilon n}{1+\delta}\right)$. Now, we consider two regimes depending on whether $\operatorname{Tr} D_j D_{j'}$ is less than or greater than $\varepsilon n$.

In the regime where $\operatorname{Tr} D_j D_{j'} \leq \varepsilon n$, we choose $t_1 = \varepsilon n^{3/4}$, yielding:

$$\frac{|x_i^T D_j D_{j'} x_{i'}|}{\|D_j x_i\|_2 \|D_{j'} x_{i'}\|_2} \leq \frac{1}{(1-\delta)n^{1/4}}$$

with probability at least $1 - 2\exp\left(\frac{-c\varepsilon n^{3/4}}{1+n^{1/4}}\right) - 4\exp\left(\frac{-c\delta^2 \varepsilon n}{1+\delta}\right)$.

In the regime where $\operatorname{Tr} D_j D_{j'} > \varepsilon n$, we choose $t_1 = \delta n^{-1/4} \operatorname{Tr} D_j D_{j'}$. Combining this with the implication of Assumption 2 that $\operatorname{Tr} D_j D_{j'} \leq (1-\gamma)n$, we have

$$\frac{|x_i^T D_j D_{j'} x_{i'}|}{\|D_j x_i\|_2 \|D_{j'} x_{i'}\|_2} \leq \frac{\delta n^{-1/4} \operatorname{Tr} D_j D_{j'}}{(1-\delta)\varepsilon n} \leq \frac{\delta(1-\gamma)}{(1-\delta)\varepsilon n^{1/4}}$$

with probability at least $1 - 2\exp\left(\frac{-c\delta^2 n^{3/4}\varepsilon}{n^{1/4}+\delta}\right) - 4\exp\left(\frac{-c\delta^2 \varepsilon n}{1+\delta}\right)$. For all $\delta \in (0,1)$, $1 - 2\exp\left(\frac{-c\delta^2 n^{3/4}\varepsilon}{n^{1/4}+\delta}\right) - 4\exp\left(\frac{-c\delta^2 \varepsilon n}{1+\delta}\right) \leq 1 - 2\exp\left(\frac{-c\varepsilon n^{3/4}}{1+n^{1/4}}\right) - 4\exp\left(\frac{-c\delta^2 \varepsilon n}{1+\delta}\right)$, and for $\delta \leq \frac{\varepsilon}{1-\gamma}$ we have $\frac{1}{(1-\delta)n^{1/4}} \geq \frac{\delta(1-\gamma)}{(1-\delta)\varepsilon n^{1/4}}$. Combining these, we have that

$$\frac{|x_i^T D_j D_{j'} x_{i'}|}{\|D_j x_i\|_2 \|D_{j'} x_{i'}\|_2} \leq \frac{1}{(1-\delta)n^{1/4}}$$

holds with probability at least $1 - 2\exp\left(\frac{-c\delta^2 n^{3/4}\varepsilon}{n^{1/4}+\delta}\right) - 4\exp\left(\frac{-c\delta^2 \varepsilon n}{1+\delta}\right)$, for all $\delta \in (0, \frac{\varepsilon}{1-\gamma})$ and all entries of a block submatrix $(A_S^T A_S)_{i,i'}$ that is off the diagonal of $A_S^T A_S$.

Now that we have high probability (and in some cases deterministic) upper and lower bounds on each entry of $A_S^T A_S$, we combine them into high probability bounds on the eigenvalues of $A_S^T A_S$. We can decompose $A_S^T A_S = B + C$, where $B$ is block diagonal and $C$ is dense except for having zeros in block-diagonal entries. First, consider a single block submatrix $B_{i,i}$ on the diagonal of $B$. This block submatrix has diagonal values deterministically 1, and off-diagonal entries bounded deterministically from below by 0 and bounded above by $\frac{(1+\delta)\sqrt{1-\gamma}}{1-\delta}$ with probability at least $1 - 2\exp\left(\frac{-c\delta^2 \varepsilon \sqrt{n}}{1+\delta}\right) - 4\exp\left(\frac{-c\delta^2 \varepsilon n}{1+\delta}\right)$. We use the variational definition of the minimum and maximum eigenvalues, and refer to $B_{i,i}$ as $\tilde{B}$ so that subscripts may denote indices within the block submatrix:

$$\lambda_{\min}(\tilde{B}) = \min_{\|u\|_2 = 1} u^T \tilde{B} u$$

$$= \min_{\|u\|_2 = 1} \sum_i \tilde{B}_{i,i} u_i^2 + \sum_{i \neq j} B_{i,j} u_i u_j$$

$$\overset{(a)}{=} 1 + \min_{\|u\|_2 = 1} \sum_{i \neq j} B_{i,j} u_i u_j$$

$$\overset{(b)}{\geq} 1 - \frac{(1+\delta)\sqrt{1-\gamma}}{1-\delta},$$

where in (a) we use the deterministic facts that diagonal entries of $\tilde{B}$ take value 1 and that $\|u\|_2 = 1$, and in (b) we use the high-probability upper bound on the magnitude of $\tilde{B}_{i,j}$ and the observation that the minimum is achieved by a vector $u \perp \mathbf{1}$ (this makes the cross terms most negative). By a similar line of reasoning, we can bound

$$\lambda_{\max}(\tilde{B}) = 1 + \max_{\|u\|_2=1} \sum_{i \neq j} B_{i,j} u_i u_j$$

$$\overset{(a)}{\leq} 1 + \frac{(1+\delta)\sqrt{1-\gamma}}{1-\delta}(s-1),$$

where in (a) we use the high-probability upper bound on the magnitude of $\tilde{B}_{i,j}$, the observation that the maximum is achieved by a vector $u \parallel \mathbf{1}$, and the requirement that the maximum dimension of $\tilde{B}$ is $s \times s$. Both of these bounds hold with probability at least $1 - 2s(s-1)\exp\left(\frac{-c\delta^2 \varepsilon \sqrt{n}}{1+\delta}\right) - 4s(s-1)\exp\left(\frac{-c\delta^2 \varepsilon n}{1+\delta}\right)$, by a union bound over all off-diagonal entries in $\tilde{B}$. The spectrum of the full block matrix $B$ is bounded by the minimum and maximum eigenvalues of its largest block, which can have size at most $s \times s$. Thus the bounds above on $\lambda_{\min}(\tilde{B})$ and $\lambda_{\max}(\tilde{B})$ also apply to $B$.

Since $A_S^T A_S = B + C$, it remains to bound the spectrum of $C$ and combine the results. The structure of $C$ is dense, with block diagonal submatrices of value zero and each other entry bounded between $-\frac{1}{(1-\delta)n^{1/4}}$ and $\frac{1}{(1-\delta)n^{1/4}}$ with probability at least $1 - 2\exp\left(\frac{-c\delta^2 n^{3/4}\varepsilon}{n^{1/4}+\delta}\right) - 4\exp\left(\frac{-c\delta^2 \varepsilon n}{1+\delta}\right)$, for all $\delta \in (0, \frac{\varepsilon}{1-\gamma})$. For this matrix, we use a coarse bound that $\|C\|_{op} \leq \frac{s}{(1-\delta)n^{1/4}}$, which holds with probability at least $1 - 2s(s-1)\exp\left(\frac{-c\delta^2 n^{3/4}\varepsilon}{n^{1/4}+\delta}\right) - 4s(s-1)\exp\left(\frac{-c\delta^2 \varepsilon n}{1+\delta}\right)$ for all $\delta \in (0, \frac{\varepsilon}{1-\gamma})$, following a union bound.

Combining these spectral bounds on $B$ and $C$ via Weyl's inequality, we have that

$$\lambda_{\min}(A_S^T A_S) \geq 1 - \frac{(1+\delta)\sqrt{1-\gamma}}{1-\delta} - \frac{s}{(1-\delta)n^{1/4}}$$

and

$$\lambda_{\max}(A_S^T A_S) \leq 1 + \frac{(s-1)(1+\delta)\sqrt{1-\gamma}}{1-\delta} + \frac{s}{(1-\delta)n^{1/4}}$$

for any $\delta \in (0, \frac{\varepsilon}{1-\gamma})$ with probability at least $1 - 2s(s-1)\exp\left(\frac{-c\delta^2 \varepsilon \sqrt{n}}{1+\delta}\right) - 2s(s-1)\exp\left(\frac{-c\delta^2 n^{3/4}\varepsilon}{n^{1/4}+\delta}\right) - 8s(s-1)\exp\left(\frac{-c\delta^2 \varepsilon n}{1+\delta}\right)$. $\qquad\square$

**Theorem 2** (Hanson-Wright (Boucheron et al., 2013)). *Let $x$ be a random vector with i.i.d. zero-mean 1-sub-Gaussian entries. Let $H$ be a square matrix. Then for a universal constant $c$*

$$\mathbb{P}\big[\,\big|x^T H x - \mathbb{E}[x^T H x]\big| \geq t\,\big] \leq 2\exp\left(-\frac{ct^2}{\|H\|_F^2 + \|H\|_{op} t}\right).$$

*If $H$ is a diagonal matrix with all diagonal entries equal to either zero or one, $\|H\|_F^2 = \operatorname{Tr} H$ and $\|H\|_{op} = 1$. We also use an asymmetric version of Hanson-Wright, derived as follows. Let*

$$H = \begin{bmatrix} 0 & \tilde{H} \\ 0 & 0 \end{bmatrix}; \quad x = \begin{bmatrix} u \\ v \end{bmatrix};$$

*yielding*

$$\mathbb{P}\left[\,\left|u^T \tilde{H} v - \mathbb{E}[u^T \tilde{H} v]\right| \geq t\right] \leq 2\exp\left(-\frac{ct^2}{\|\tilde{H}\|_F^2 + \|\tilde{H}\|_{op} t}\right).$$

## D   PROOF OF THEOREM 1

*Proof.* The proof of Theorem 1 combines Lemma 1 with Theorem 3 (Theorem 1 in Jain et al. (2014)), which shows that IHT with an inflated sparsity level can recover a sparse signal in a linear inverse problem as long as the sensing matrix satisfies the restricted strong convexity and restricted

strong smoothness properties with any positive finite parameters; i.e. with an arbitrary finite restricted condition number.

Theorem 3 shows directly that, under the conditions in the theorem statement, the objective value converges as $f(w^K) - f(w^\star) \leq \epsilon$. This implies convergence of iterates due to restricted strong convexity and the fact that $\nabla f(w^\star) = 0$:

$$f(w^K) - f(w^\star) \geq \langle w^K - w^\star, \nabla f(w^\star) \rangle + \frac{\alpha}{2} \left\| w^K - w^\star \right\|_2^2 = \frac{\alpha}{2} \left\| w^K - w^\star \right\|_2^2,$$

which proves the additional result that $\left\| w^K - w^\star \right\|_2^2 \leq \frac{2}{\alpha} \epsilon$.

We note that the proof in Jain et al. (2014) also implies that, if IHT projects onto the smaller sparsity level $s$ rather than the inflated sparsity level $\tilde{s}$, each step of IHT is still guaranteed to not increase the MSE loss $f(w)$; the requirement that $\tilde{s} > s$ allows for strict objective decrease in each step. $\square$

**Theorem 3** (Jain et al. (2014)). *Assume that the objective $f$ has restricted strong convexity parameter $\alpha$ and restricted strong smoothness parameter $L$ at sparsity level $2\tilde{s} + s$, with $\tilde{s} > 32\left(\frac{L}{\alpha}\right)^2 s$. Assume that $\theta^\star = \arg\min_{\|\theta\|_0 \leq s} f(\theta)$, i.e. that the true signal is $s$-sparse. Then IHT with projection (hard thresholding) to sparsity level $\tilde{s}$ and step size $\eta = \frac{2}{3L}$, run for $K = \mathcal{O}\left(\frac{L}{\alpha} \log\left(\frac{f(\theta^0)}{\epsilon}\right)\right)$ iterations, achieves*

$$f(\theta^K) - f(\theta^\star) \leq \epsilon.$$

# E  PROOF THAT ASSUMPTION 2 FOLLOWS FROM ASSUMPTION 1 WITH HIGH PROBABILITY

Let $D_i = \text{diag}\left(\mathbb{I}\{Xh_i \geq 0\}\right) \in \mathbb{R}^{n \times n}$, with $\{D_i\}_{i=1}^p$ as the set of all such distinct activation patterns possible under Assumption 1 with data $X \in \mathbb{R}^{n \times d}$, whose entries are drawn i.i.d. $\sim \mathcal{N}(0,1)$. Assumption 2 has the following two components:

1. $\text{Tr}\, D_i \geq \varepsilon n$ for all $i \in [p]$, for some $\varepsilon \in (0,1)$.
2. For all $i \neq i'$, the diagonals of $D_i$ and $D_{i'}$ differ in at least $\gamma n$ positions, for some $\gamma \in (0,1)$.

## E.1  COMPONENT 1: LOWER BOUND ON TRACE OF ACTIVATION PATTERNS $D_i$

Component 1 follows from Lemma 2, which does not require Assumption 1.

**Lemma 2** (based on Ergen et al. (2019)). *Let $S = \{i : x_i^T h > 0\}$, where $x_i$ are i.i.d standard Gaussian vectors distributed as $\mathcal{N}(0, I_d)$. Then with probability at least $1 - e^{-n\left(\varphi(1-\varepsilon) - \mathcal{H}(\varepsilon)\right)}$, $\inf_h |S| \geq n\varepsilon$. Here $\varepsilon \in (0,1)$, $\varphi$ is a fixed numerical constant satisfying $\frac{1}{2} - \sqrt{8\varphi} > 0$, $n$ satisfies $n\left(\frac{1}{2} - \sqrt{8\varphi}\right) \geq d$, and $\mathcal{H}$ is the binary entropy function.*

*Proof.* Consider the symmetric event $E := \sup_{h \neq 0} |\{i : x_i^T h \leq 0\}| \geq n(1 - \varepsilon)$. Then

$$\mathbb{P}[E] \leq \sum_{\substack{V \subseteq [n] \\ |V| \geq n(1-\varepsilon)}} \mathbb{P}\left[\exists h \neq 0 \text{ s.t. } x_i^T h \leq 0, \forall i \in V\right]$$

$$\leq \binom{n}{n(1-\varepsilon)} e^{-\varphi n(1-\varepsilon)}$$

$$\leq e^{-n\left(\varphi(1-\varepsilon) - \mathcal{H}(\varepsilon)\right)}$$

in which the second inequality follows from the Kinematic Formula (by flipping the sign of $h$). $\square$

**Theorem 4** (Kinematic Formula (Amelunxen et al., 2014)). *Let $X$ be an $n \times d$ i.i.d. Gaussian matrix and $G = X\Sigma^{1/2}$ with any $\Sigma \succ 0$. If $n$ satisfies $n\left(\frac{1}{2} - \sqrt{8\varphi}\right) \geq d$, we have*

$$\mathbb{P}\left[\exists h \neq 0 \text{ s.t. } Gh \geq 0\right] = \mathbb{P}\left[\exists \tilde{h} \neq 0 \text{ s.t. } X\tilde{h} \geq 0\right] \leq e^{-\varphi n}.$$

**Remark 3.** *If we further assume Assumption 1, specifically that $\|h\|_0 \le s_i \le k$, we can tighten Lemma 2 as follows. If $\|h\|_0 \le k \le d$, and we set $\varphi = \frac{1}{128}$, then $\operatorname{Tr} D_i \ge \varepsilon n$ with probability at least $1 - e^{-n\left(\frac{1-\varepsilon}{128} - \mathcal{H}(\varepsilon)\right)}$ as long as $n \ge 4k$. Following a union bound over $p$ activation patterns, $\operatorname{Tr} D_i \ge \varepsilon n$ for all $i \in [p]$ with probability at least $1 - pe^{-n\left(\frac{1-\varepsilon}{128} - \mathcal{H}(\varepsilon)\right)}$, as long as $n \ge 4k$.*

### E.2 COMPONENT 2: HAMMING SEPARATION OF ACTIVATION PATTERNS

**Definition 1** ($\delta$-isometric embedding (Plan & Vershynin, 2014))**.** *A map $f : X \to Y$ is a $\delta$-isometry between metric space $X$ with distance metric $d_X$ and metric space $Y$ with distance metric $d_Y$ if, for all $x, x' \in X$, $|d_Y(f(x), f(x')) - d_X(x, x')| \le \delta$.*

**Theorem 5** (Hamming embedding, Theorem 1.5 in Plan & Vershynin (2014))**.** *Consider a subset $K \subseteq S^{d-1}$ and let $\delta > 0$. Let $X$ be an $n \times d$ random matrix with independent $\mathcal{N}(0,1)$ entries. Let $n \ge C\delta^{-6}w(K)^2$, where $w(K) := \mathbb{E}\sup_{x \in K} |\langle g, x \rangle|$ is the Gaussian mean width of $K$, with $g \sim \mathcal{N}(0, I_d)$. Then with probability at least $1 - 2\exp(-c\delta^2 n)$, the sign map $f(x) = \operatorname{sign}(Xx)$, $f : K \to \{-1, 1\}^n$ is a $\delta$-isometric embedding between $K \subseteq S^{d-1}$ with normalized geodesic distance metric $d_G(x, x') = \frac{1}{\pi}\cos^{-1}(x^T x')$ and $\{-1, 1\}^n$ with normalized Hamming distance metric $d_H(f(x), f(x')) = \frac{1}{n}\sum_{i=1}^n f(x)_i \ne f(x')_i$. Here $C$ and $c$ denote positive absolute constants.*

**Corollary 1.** *Theorem 5 may be restated so as to apply to unnormalized generator vectors from $K \subseteq \mathbb{R}^d$ and indicator-based rather than sign-based activation pattern embedding. Consider a subset $K \subseteq \mathbb{R}^d$ and let $\delta > 0$. Let $X$ be an $n \times d$ random matrix with independent $\mathcal{N}(0,1)$ entries. Let $n \ge C\delta^{-6}w(K)^2$, where $w(K) := \mathbb{E}\sup_{x \in K} |\langle g, \frac{x}{\|x\|_2}\rangle|$ is the normalized Gaussian mean width of $K$, with $g \sim \mathcal{N}(0, I_d)$. Then with probability at least $1 - 2\exp(-c\delta^2 n)$, the indicator map $f(x) = \mathbb{I}\{Xx \ge 0\}$, $f : K \to \{0, 1\}^n$ is a $\delta$-isometric embedding between $K \subseteq \mathbb{R}^d$ with normalized geodesic distance metric $d_G(x, x') = \frac{1}{\pi}\cos^{-1}\left(\frac{x^T x'}{\|x\|_2\|x'\|_2}\right)$ and $\{0, 1\}^n$ with normalized Hamming distance metric $d_H(f(x), f(x')) = \frac{1}{n}\sum_{i=1}^n f(x)_i \ne f(x')_i$. Here $C$ and $c$ denote positive absolute constants.*

Corollary 1 ensures that a set of generator vectors $\{h_i\}_{i=1}^p$ that are sufficiently separated in normalized geodesic distance will yield activation patterns $D_i = \operatorname{diag}(\mathbb{I}\{Xh_i \ge 0\})$ whose diagonals are separated in normalized Hamming distance, with high probability for i.i.d. Gaussian data $X \in \mathbb{R}^{n \times d}$. Specifically, for the diagonals of $D_i$ and $D_{i'}$ to differ in at least $\gamma n$ positions for all $i \ne i'$ with probability at least $1 - 2\exp(-c\delta^2 n)$, we require a set of generator vectors $\{h_i\}_{i=1}^p$ that (1) include the planted first layer weights $u_i^\star$, and (2) are separated by at least $\gamma + \delta$ in normalized geodesic distance. In Appendix E.3 we show that both of these properties hold with high probability for both of the sparse weight conditions in Assumption 1 (recall that $u_i^\star$ are the first layer weights and $v_i^\star$ are the second layer weights, which are fused during IHT):

(a) $u_i^\star \in \{-1, 0, 1\}^d$, $\|u_i^\star\|_0 = k$, $v_i^\star \in \mathbb{R} \ \forall i \in [p]$ and $kp \le s$, or

(b) $u_i^\star \in \mathbb{R}^d$, $\|u_i^\star\|_0 = s_i \in [s_{\min}, k]$, $v_i^\star \in \{-1, 1\} \ \forall i \in [p]$ and $\sum_{i=1}^p s_i \le s$ holds.

### E.3 SAMPLING SPARSE ARRANGEMENTS

#### E.3.1 REAL-VALUED PLANTED NEURONS

We now show that a random sampling of hyperplane arrangements can be guaranteed to contain the planted activation patterns, while simultaneously ensuring a packing of the Euclidean sphere in $\mathbb{R}^d$.

**Theorem 6.** *Let $X \in \mathbb{R}^{n \times d}$ have i.i.d. $\mathcal{N}(0,1)$ entries. Fix $m$ unknown vectors $w_1, \ldots, w_m \in \mathbb{R}^d$ each with $\|w_i\|_0 = s_i \in [s_{min}, k]$, and an error tolerance $0 < \tilde{\epsilon} < 1$. Note that the choice of $\tilde{\epsilon}$ affects the permissible sparsity range $[s_{min}, k]$. Set*

$$T = \left(\frac{\log(2n)}{c}\right)^k \log\left(\frac{2m}{\tilde{\epsilon}}\right),$$

*where $c > 0$ is an absolute constant. Consider the set of all supports $S$ with $|S| \in [s_{min}, k]$, and draw $T$ supports from this set uniformly at random. For each randomly drawn support $S$, draw $|S|$*

*values i.i.d. from $\mathcal{N}(0,1)$ and embed these in $\mathbb{R}^d$ by setting entries in $S$ to their random Gaussian values and zero–padding outside $S$. Record the two collections*

$$\Gamma = \big\{\mathbb{I}\big[Xh_j \geq 0\big] : 1 \leq j \leq T\big\}, \qquad G = \big\{\tfrac{\widetilde{h}_j}{\|\widetilde{h}_j\|_2} : 1 \leq j \leq T\big\},$$

*where $\widetilde{h}_j$ denotes the zero–padded generator and $h_j$ is its normalized version. There exists $\delta > 0$ such that, with probability at least $1 - \tilde{\epsilon}$ over the draws of $X$ and all $T$ generators, the following hold simultaneously:*

1. **Coverage:** $\mathbb{I}\big[Xw_i \geq 0\big] \in \Gamma$ *for every $i \in \{1, \dots, m\}$.*

2. **Minimum geodesic separation:** *For all distinct $g, g'$ in $G$, $\|g - g'\|_2 \geq \tilde{\delta}$. This Euclidean separation of unit vectors implies geodesic separation: $d_G(g, g') = \frac{1}{\pi}\cos^{-1}(g^T g') \geq \frac{0.69}{\pi}\tilde{\delta}^2$.*

*Proof.* The strategy used to prove coverage is to show that the cones $\{u : \mathrm{sign}(Xu) = \mathrm{sign}(Xh)\}$ are not too narrow, for Gaussian i.i.d. training data $X \in \mathbb{R}^{n \times d}$ and a fixed vector $h \in \mathbb{R}^d$. Specifically, a bound on the cone sharpness developed in Kim & Pilanci (2024) implies that the probability that a uniformly sampled vector on the sphere falls into this cone is at least $O\big((\log n)^{-d}\big)$. We then apply this result to $n \times s_i$ submatrices of $X$ to translate it to sparse generators, and control the error probability via the union bound. We first reintroduce the notion of cone sharpness:

*Cone sharpness.* For any support $S$ and non-zero $u \in \mathbb{R}^{s_i}$, set $D(u) := \mathrm{diag}\big(\mathbb{I}[X_S u \geq 0]\big)$ and define the cone $\mathcal{K}_S(u) := \{v \in \mathbb{R}^{s_i} : (2D(u) - I)X_S v \geq 0\}$. By the cone–sharpness bound of Kim & Pilanci (2024) there are universal constants $c, c_1 > 0$ such that

$$\mathbb{P}_X\Big[C\big(\mathcal{K}_S(u), \tfrac{u}{\|u\|_2}\big) \leq C_*\Big] \geq 1 - \tilde{\delta}_{s_i}, \quad C_* := 2 + 200c\sqrt{c\log(2n)}, \quad \tilde{\delta}_{s_i} := n^{-10} + e^{-c_1 s_i},$$

where the sharpness $C(\mathcal{K}, z)$ of a cone $\mathcal{K}$ with respect to a fixed unit vector $z$ is defined as $C(\mathcal{K}, z) := \min_{u,v \in \mathcal{K},\ u-v=z} \|u\|_2 + \|v\|_2$ following Kim & Pilanci (2024). Let $\mathcal{E}$ be the high probability event that the cone sharpness for each of the $m$ planted neurons is at most $C_*$; $\mathcal{E}$ occurs with probability at least $1 - m(n^{-10} + e^{-c_1 s_{\min}})$. Now we relate cone sharpness to the probability of sampling a specific pattern.

*Spherical cap inclusion.* Fix $S$ and $u \neq 0$ and write $z := u/\|u\|_2$. On $\mathcal{E}$ there exist $a, b \in \mathcal{K}_S(u)$ with $a - b = z$ and $\|a\|_2 + \|b\|_2 \leq C_*$. Setting $q := (a+b)/2$ yields $\langle q, z \rangle \geq 1/(2C_*)$. Therefore the spherical cap

$$\mathcal{C}_z := \big\{y \in \mathbb{S}^{|S|-1} : \langle y, z \rangle \geq 1/(2C_*)\big\}$$

is contained in $\mathcal{K}_S(u)$.

*Cap measure.* For $h \sim \mathcal{N}(0, I_{s_i})$ the direction $h/\|h\|_2$ is uniform on $\mathbb{S}^{s_i-1}$. Standard surface–measure estimates give

$$p_{s_i} := \mathbb{P}\big[\mathbb{I}[X_S h \geq 0] = \mathbb{I}[X_S u \geq 0]\big] \geq \mathrm{Surf}_{s_i-1}(\mathcal{C}_z) \geq \frac{c^{s_i}}{\big(\log(2n)\big)^{s_i}}.$$

*Coverage probability.* As a consequence of the above inequality, the arrangement pattern of each planted neuron is sampled with probability at least $\frac{c^{s_i}}{\big(\log(2n)\big)^{s_i}} \geq \Big(\frac{c}{\log(2n)}\Big)^k$. After $T$ samples, the probability that we have not yet sampled all $m$ planted neuron activation patterns is at most $m\left(1 - \Big(\frac{c}{\log(2n)}\Big)^k\right)^T$; after $T = \Big(\frac{\log(2n)}{c}\Big)^k \log\big(\frac{2m}{\tilde{\epsilon}}\big)$ random draws we are guaranteed to sample all $m$ planted patterns with probability at least $1 - \frac{\tilde{\epsilon}}{2}$.

*Packing of generators.* Consider any two generator vectors $h, h'$ with supports $S, S'$, respectively. We have

$$h^T h' = \frac{\sum_{i \in S \cap S'} \tilde{h}_i \tilde{h}_i'}{\sqrt{\sum_{i \in S} \tilde{h}_i^2}\sqrt{\sum_{i \in S'} \tilde{h}_i'^2}},$$

where $\tilde{h}_i$ and $\tilde{h}'_i$ are i.i.d. distributed as $\mathcal{N}(0,1)$. Using a union bound over asymmetric Hanson-Wright (Theorem 2) in the numerator and symmetric Hanson-Wright (twice) in the denominator, we have

$$\mathbb{P}\left[|h^T h'| \geq \frac{t_1}{\sqrt{|S|+t_2}\sqrt{|S'|+t_3}}\right] \leq 2\exp\left(\frac{-ct_1^2}{|S\cap S'|+t_1}\right) + 2\exp\left(\frac{-ct_2^2}{|S|+t_2}\right) + 2\exp\left(\frac{-ct_3^2}{|S'|+t_3}\right).$$

We choose $t_2 = \gamma|S|$, $t_3 = \gamma|S'|$, yielding

$$\mathbb{P}\left[|h^T h'| \geq \frac{t_1}{(1+\gamma)\sqrt{|S||S'|}}\right] \leq 2\exp\left(\frac{-ct_1^2}{|S\cap S'|+t_1}\right) + 2\exp\left(\frac{-c\gamma^2|S|}{1+\gamma}\right) + 2\exp\left(\frac{-c\gamma^2|S'|}{1+\gamma}\right).$$

Next, we choose $t_1 = \left(1 - \frac{\tilde{\delta}^2}{2}\right)(\gamma+1)\sqrt{|S||S'|}$ to yield

$$\mathbb{P}\left[|h^T h'| \geq 1 - \frac{\tilde{\delta}^2}{2}\right] \leq 2\exp\left(\frac{-c\left(1-\frac{\tilde{\delta}^2}{2}\right)^2(\gamma+1)^2|S||S'|}{|S\cap S'| + \left(1-\frac{\tilde{\delta}^2}{2}\right)(\gamma+1)\sqrt{|S||S'|}}\right)$$
$$+ 2\exp\left(\frac{-c\gamma^2|S|}{1+\gamma}\right) + 2\exp\left(\frac{-c\gamma^2|S'|}{1+\gamma}\right).$$

Assuming that all planted neurons, and thus all generator vectors we need to consider, have sparsity level $s_i \in [s_{\min}, k]$, we have

$$\mathbb{P}\left[|h^T h'| \geq 1 - \frac{\tilde{\delta}^2}{2}\right] \leq 2\exp\left(\frac{-c\left(1-\frac{\tilde{\delta}^2}{2}\right)^2(\gamma+1)^2 s_{\min}^2}{k + \left(1-\frac{\tilde{\delta}^2}{2}\right)(\gamma+1)k}\right) + 4\exp\left(\frac{-c\gamma^2 s_{\min}}{1+\gamma}\right).$$

For sufficiently large $s_{\min}$, a union bound over all $\binom{T}{2}$ pairs of generators allows us to bound $\mathbb{P}\left[|h^T h'| \geq 1 - \frac{\tilde{\delta}^2}{2}\right] \leq \frac{\tilde{\epsilon}}{2} - m(n^{-10} + e^{-c_1 s_{\min}})$ uniformly over all pairs $h, h'$, as required for an overall failure probability at most $\tilde{\epsilon}$. Finally, the Euclidean packing follows as

$$\|h - h'\|_2^2 = \|h\|_2^2 + \|h'\|_2^2 + 2h^T h' = 2 + 2h^T h' \geq 2 - (2 - \tilde{\delta}^2) = \tilde{\delta}^2.$$

Euclidean $\tilde{\delta}$-packing implies the stated separation in normalized geodesic distance as follows:

$$d_G(g, g') = \frac{1}{\pi}\cos^{-1}\left(\frac{g^T g'}{\|g\|_2 \|g'\|_2}\right)$$
$$\overset{(a)}{\geq} \frac{1.38}{\pi}(1 - g^T g')$$
$$\overset{(b)}{\geq} \frac{0.69}{\pi}\tilde{\delta}^2,$$

where in (a) we use the fact that $\cos^{-1}(1-x) \geq 1.38x$ for $x \in [0,1]$ and in (b) we use that $\|g\|_2 = \|g'\|_2 = 1$ and $\|g - g'\|_2^2 = \|g\|_2^2 + \|g'\|_2^2 - 2g^T g' = 2 - 2g^T g \geq \tilde{\delta}^2$.

$\square$

### E.3.2 DISCRETE-VALUED PLANTED NEURONS

**Theorem 7.** *Fix integers $d$ and $k \leq d$. For a subset $S \subseteq [d]$ with entries $S_j$ and $|S| = k$, define the generator set*

$$\mathcal{G}_S := \left\{g(\sigma) \in \{-1,0,1\}^d \;:\; \sigma \in \{-1,1\}^k,\; g(\sigma)_{S_j} = \sigma_j,\; g(\sigma)_\ell = 0 \text{ if } \ell \notin S\right\},$$

*i.e. all possible sign assignments on the coordinates in $S$ and zeros elsewhere ($|\mathcal{G}_S| = 2^k$). Given a matrix $X \in \mathbb{R}^{n \times d}$ form the associated* arrangement list

$$\Gamma_S := \left\{\mathbb{I}[Xg \geq 0] \;:\; g \in \mathcal{G}_S\right\} \subseteq \{0,1\}^n.$$

(i) **Coverage.** *Let $w_1, \ldots, w_m \in \{-1, 0, 1\}^d$ be $k$-sparse vectors (each with exactly $k$ non-zeros). Then for every $i \in \{1, \ldots, m\}$*

$$\mathbb{I}[Xw_i \geq 0] \in \Gamma_{supp(w_i)}.$$

(ii) **Minimum geodesic separation.** *For any two distinct $k$-sparse vectors $u, v \in \{-1, 0, 1\}^d$,*

$$d_G(u, v) = \frac{1}{\pi} \cos^{-1}\left(\frac{u^T v}{\|u\|_2 \|v\|_2}\right) \geq \frac{0.69}{\pi k}.$$

*Proof. (i) Coverage.* Fix $i \in \{1, \ldots, m\}$ and set $S = \text{supp}(w_i)$. Because $w_i$ has entries $\pm 1$ on $S$ and zeros elsewhere, there exists $\sigma \in \{-1, 1\}^{|S|}$ such that $w_i = g(\sigma) \in \mathcal{G}_S$. Hence the pattern $\mathbf{1}[Xw_i \geq 0]$ belongs to $\Gamma_S$.

*(ii) Separation.* Let $u \neq v$ be $k$-sparse vectors in $\{-1, 0, 1\}^d$. There is an index $j$ with $u_j \neq v_j$, so $|u_j - v_j| \geq 1$. Therefore $\|u - v\|_2^2 = \sum_{\ell=1}^d (u_\ell - v_\ell)^2 \geq (u_j - v_j)^2 \geq 1$, implying $\|u - v\|_2 \geq 1$.

From Euclidean separation we can infer normalized geodesic separation as follows:

$$
\begin{aligned}
d_G(u, v) &= \frac{1}{\pi} \cos^{-1}\left(\frac{u^T v}{\|u\|_2 \|v\|_2}\right) \\
&\overset{(a)}{\geq} \frac{1.38}{\pi}\left(1 - \frac{u^T v}{\|u\|_2 \|v\|_2}\right) \\
&\overset{(b)}{\geq} \frac{1.38}{\pi}\left(1 - \frac{\|u\|_2^2 + \|v\|_2^2 - 1}{2 \|u\|_2 \|v\|_2}\right) \\
&\overset{(c)}{\geq} \frac{0.69}{\pi k},
\end{aligned}
$$

where in (a) we use the fact that $\cos^{-1}(1 - x) \geq 1.38x$ for $x \in [0, 1]$, in (b) we use that $\|u - v\|_2^2 = \|u\|_2^2 + \|v\|_2^2 - 2u^T v \geq 1$, and in (c) we use that $\|u\|_2^2 = \|v\|_2^2 = k$ since both $u$ and $v$ have exactly $k$ nonzero entries each with magnitude one. $\square$

