# OpenReview forum: "A Recovery Guarantee for Sparse Neural Networks"
_ICLR.cc/2026/Conference — ICLR 2026 Poster_

### Official Review · Reviewer_T4on · 2025-10-26

**Soundness:** 3
**Presentation:** 3
**Contribution:** 3
**Rating:** 6
**Confidence:** 3

**Summary:**

The paper studies sparse training through a recovery-guarantee lens for two-layer ReLU MLPs with scalar output. Using a convex reformulation that enumerates activation patterns and fuses layer weights, the training objective becomes a structured linear sensing problem. Under Gaussian inputs and two separability conditions on activation patterns, the sensing matrix satisfies restricted strong convexity and smoothness, which makes the planted sparse weights uniquely identifiable. Experiments on planted MLP recovery, MNIST classification, and implicit neural representations show the proposed method is competitive with or better than iterative magnitude pruning while using far less memory, even beyond the exact theory regime.

**Strengths:**

1.The paper gives identifiability and efficient recovery for planted sparse weights in two-layer scalar-output MLPs under Gaussian data.
2. The activation-pattern convexification connects ReLU training to linear sensing in a way that enables analysis with classical tools.
3. Across planted recovery, MNIST, and image INRs, IHT is typically stronger than IMP while using memory that scales with s rather than with dense parameter counts.

**Weaknesses:**

1. Scope limited to shallow scalar-output MLPs in theory.
2. The main theorem relies on i.i.d. Gaussian covariates and full enumeration of activation patterns, which can be unrealistic or computationally heavy.

**Questions:**

1. Extension beyond Gaussian inputs? Can the proof be adapted using sub-Gaussian or whitened real data, and what additional conditions would be needed on X?

2. The experiments use randomly sampled patterns and sequential convex updates. Is there a provable guarantee when A is updated during training, perhaps via a stability or tracking argument?

3. The appendix describes vector-output handling and layer-wise training. Could the convex deep-net formulations be combined with your analysis to yield recovery guarantees for deeper or multi-output networks?

---

> ### Author Response · Authors · 2025-11-19
>
> We thank the reviewer for their thoughtful and positive engagement with our work. We are happy to respond to the reviewer’s comments and questions below, and to answer any further questions that may arise during the discussion period.
>
> >Extension beyond Gaussian data in the theory:
>
> It should be possible to extend our proof to sub-Gaussian or whitened covariates $X$, though the extension is nontrivial because it would affect the distribution of activation patterns. This requires an involved analysis of the chambers of hyperplane arrangements resulting from sub-Gaussian data. However, for rotationally invariant distributions, this analysis becomes more tractable. We leave these questions for follow-up work due to the complexities in the probabilistic analysis.
>
> >Extension to randomly sampled patterns and sequential convex updates, instead of requiring full pattern enumeration in the theory:
>
> We agree that this extension is a high priority for future work. Empirically this approach shows strong performance, but proving exact recovery with random pattern initialization and sequential convex updates is a challenging endeavor that is beyond the scope of the present paper, which focuses on providing a first sparse recovery guarantee for MLPs.
>
> >Extension to deeper MLPs or vector-valued output in the theory:
>
> As above, we completely agree that deeper and vector-valued MLPs are exciting directions for follow-up research. Extension to deeper networks should be possible given recent convex reformulations of deep MLPs [1] under path regularization; we expect the primary challenge in such an extension would be enumerating over a much larger search space of possible multi-layer activation patterns. For the present paper, we show experimentally that IHT achieves competitive performance for training sparse 3-layer networks following a layerwise optimization strategy. This result is encouraging, though we expect even better performance could be possible by formulating a bespoke sparse IHT for deeper networks (instead of layerwise training).
>
> Regarding vector-valued outputs, empirically the primary challenge lies in step size selection (we have a closed form optimal stepsize in the scalar-output case, but currently our vector-output variant of sparse IHT requires manual step size tuning). Theoretically, the challenge lies in the restriction that the weight matrix (which is then a matrix instead of a vector, as it is in the scalar-output case) is both sparse and block-wise rank-one (see equation 6 in the appendix), instead of purely sparse. We expect that it should be possible to extend our theory to this more challenging vector-output case, but the extension is quite nontrivial and thus deferred to future work.
>
> [1] Ergen, Tolga, and Mert Pilanci. "Path regularization: A convexity and sparsity inducing regularization for parallel ReLU networks." Advances in Neural Information Processing Systems 36 (2023): 59761-59786.

---

> > ### Comment · Reviewer_T4on · 2025-11-27
> >
> > The author's response effectively addressed my main concerns. I will keep my positive score.

---

### Official Review · Reviewer_A2Ki · 2025-10-26

**Soundness:** 2
**Presentation:** 3
**Contribution:** 3
**Rating:** 6
**Confidence:** 2

**Summary:**

This paper presents the theoretical recovery guarantees for sparse ReLU neural network weights, focusing on two-layer scalar-output MLP networks. The authors prove that under certain structural conditions on sparse network weights and random Gaussian training data, an Iterative Hard Thresholding algorithm can exactly recover these weights with memory linear in the number of nonzero weights. The theoretical analysis uses convex reformulations of MLPs and establishes restricted strong convexity and restricted smoothness properties. Experimental validation on planted MLPs, MNIST classification, and implicit neural representations demonstrates competitive or superior performance compared to iterative magnitude pruning.

**Strengths:**

- This work provides formal recovery guarantees for sparse neural network weights, filling an important gap between compressed sensing theory and neural network optimization.
- The paper clearly articulates assumptions and provides detailed proofs showing how these conditions arise with high probability under Gaussian data, making the theoretical results verifiable and interpretable.
- The experimental studies, which extend beyond the theoretical analysis, demonstrate the broader applicability of the guarantees, including vector outputs, deeper networks, and various tasks.

**Weaknesses:**

- The theoretical guarantees, although interesting and important, are restrictive. It is limited to two-layer, scalar-output networks with Gaussian random data. However, real-world applications may involve deeper networks, structured data distributions, and multi-dimensional outputs.
- Assumption 1 requires concrete weight structures, i.e., binary hidden weights or binary output weights, which may not reflect realistic sparse networks.
- The paper only compares against IMP. More recent sparse training methods, such as dynamic sparse training, pruning at initialization variants, and other memory-efficient approaches, can be compared. Another limitation is that the runtime analysis, with IHT sometimes being slower than IMP for larger problems, which may be a limitation in practice.

**Questions:**

- How can the proposed analysis be extended to non-Gaussian data distributions?
- More analysis on how performance degrades when assumptions are violated would strengthen the work.

---

> ### Author Response · Authors · 2025-11-19
>
> We thank the reviewer for their thoughtful and positive engagement with our work. We are happy to respond to the reviewer’s comments and questions below, and to answer any further questions that may arise during the discussion period.
>
> >Extensions to deeper networks, structured (non-Gaussian) data, and vector-valued outputs:
>
> We agree that extensions of our theory to deeper MLPs, structured data, and vector-valued outputs are all exciting directions for future work, and likely possible given our promising experimental results in these settings.
>
> Extension to deeper networks should be possible given recent convex reformulations of deep MLPs [1] under path regularization; we expect the primary challenge in such an extension would be enumerating over a much larger search space of possible multi-layer activation patterns. For the present paper, we show experimentally that IHT achieves competitive performance for training sparse 3-layer networks following a layerwise optimization strategy. This result is encouraging, though we expect even better performance could be possible by formulating a bespoke sparse IHT for deeper networks (instead of layerwise training).
>
> We focus on Gaussian data to make the analysis tractable, though we agree that real-world images are generally not Gaussian-distributed. However, in many cases real-world structured, non-Gaussian data (such as image pixel values) is first whitened in a preprocessing step to make the covariates mimic a standard Gaussian distribution, as this is often easier for models to learn with. We hope that future work can extend our analysis to better understand what types of data correlation are benign versus problematic for sparse MLP weight recovery, but we believe that the prevalence of data whitening in practice means that even our current result on Gaussian data is not far from practical relevance.
>
> Regarding vector-valued outputs, empirically the primary challenge lies in step size selection (we have a closed form optimal stepsize in the scalar-output case, but currently our vector-output variant of sparse IHT requires manual step size tuning). Theoretically, the challenge lies in the restriction that the weight matrix (which is then a matrix instead of a vector, as it is in the scalar-output case) is both sparse and block-wise rank-one (see equation 6 in the appendix), instead of purely sparse. We expect that it should be possible to extend our theory to this more challenging vector-output case, but the extension is quite nontrivial and thus deferred to future work.
>
> [1] Ergen, Tolga, and Mert Pilanci. "Path regularization: A convexity and sparsity inducing regularization for parallel ReLU networks." Advances in Neural Information Processing Systems 36 (2023): 59761-59786.
>
> >Assumption 1 requires concrete weight structures:
>
> It is true that Assumption 1 requires either the hidden layer or output layer weights to take binary signed values. In the context of a scalar-output MLP, however, this restriction is not as restrictive as it may sound. For example, a sparse scalar-output two-layer MLP without this restriction could easily be converted into one that satisfies Assumption 1, by replacing each $u_i \leftarrow u_i |v_i|$ and each $v_i \leftarrow sign(v_i)$. Such an MLP would satisfy Assumption 1 while having exactly the same input–output behavior as the original unconstrained network. We do note, however, that in a vector-output MLP the direct extension of Assumption 1 would be more restrictive than it is in the scalar-output case. Determining an appropriately general vector-valued variant of Assumption 1 is a key step for future work to extend our results to vector-output MLPs.

---

> > ### Author Response · Authors · 2025-11-19
> >
> > >Baselines beyond IMP, and runtime limitation:
> >
> > In the time-limited rebuttal period we are unable to replicate a full comparison against these methods, but we chose to compare against iterative magnitude pruning (IMP) specifically because it is a strong baseline. We also intentionally compare against a particularly strong variant of IMP, in which only 10% of the active weights are pruned at each step (compared to the more aggressive default schedule of 20%, which has lower performance but faster training [2]). In initial comparisons we also considered global and layerwise magnitude pruning and found that IMP was consistently a stronger baseline. Moreover, in recent literature IMP has been shown as a consistently high-performing method for network pruning, with its main drawback being high memory and computational complexity (see e.g. [3] and [4], which acknowledge IMP as the highest-performing pruning method but seek computationally cheaper alternatives).
> >
> > Regarding the relative runtimes of IHT and IMP, we generally find that IHT is faster than IMP for full-batch gradient updates and shallow, scalar-output networks that correspond most closely with the setting of our theoretical analysis. We are optimistic that future work that extends our theoretical analysis to the more challenging settings of deeper and vector-output networks will also yield runtime improvements in these settings. For example, these improvements could come from a more principled step size selection mechanism (presently our scalar-output experiments use a theoretically-optimal closed form step size, whereas our vector-output experiments require manual stepsize tuning which is surely suboptimal).
> >
> > [2] Frankle, Jonathan, and Michael Carbin. "The lottery ticket hypothesis: Finding sparse, trainable neural networks." ICLR, 2019.
> >
> > [3] Gharatappeh, Soheil, and Salimeh Yasaei Sekeh. "Information Consistent Pruning: How to Efficiently Search for Sparse Networks?." International Workshop on AI for Transportation. Springer, Singapore, 2025.
> >
> > [4] Saikumar, Dhananjay, and Blesson Varghese. "DRIVE: Dual Gradient-Based Rapid Iterative Pruning." arXiv preprint arXiv:2404.03687 (2024).
> >
> > >How performance degrades when assumptions are violated:
> >
> > While our theoretical results do not include sensitivity analysis with respect to assumption violation (as this would be a substantial undertaking), our experiments provide an empirical window onto IHT performance under violated assumptions. Indeed, our experiments include real non-Gaussian data such as MNIST images and Fourier-feature-embedded pixel coordinates. Moreover, in these real-data experiments there is no guarantee of a true “planted” sparse MLP to be recovered–so Assumption 1 is surely only approximately satisfied. The strong performance of IHT in these settings suggests that its performance degrades gracefully even under assumption violation. We hope that future work will formalize this robustness, for example by proving approximate sparse recovery in cases where the true planted MLP is only approximately sparse.

---

### Official Review · Reviewer_XpEf · 2025-11-01

**Soundness:** 3
**Presentation:** 3
**Contribution:** 3
**Rating:** 6
**Confidence:** 4

**Summary:**

This paper introduces and studies a method for training sparse shallow ReLU neural networks. The method reformulates the neural network optimization problem as a convex optimization problem and uses iterative hard-thresholding to obtain a sparse solution. Theoretical results guarantee the recovery under reasonable assumptions in the case of shallow networks. Empirical experiments verify that the method also works well for deep networks.

**Strengths:**

The method for training sparse networks is innovative. The theoretical guarantees are interesting and relevant. The practical performance of the methods is good. The presentation of the paper is good, it is easy to read and clear.

**Weaknesses:**

The theory only applies to shallow networks. It would be interesting to try to extend it to deeper networks. Nonetheless, this is a valuable theoretical contribution.

**Questions:**

Can the theory be extended beyond Gaussian training data?

---

> ### Author Response · Authors · 2025-11-19
>
> We thank the reviewer for their thoughtful and positive engagement with our work. We are happy to respond to the reviewer’s comments and questions below, and to answer any further questions that may arise during the discussion period.
>
> >Extension to deeper networks:
>
> We agree that extension of our theoretical analysis to deeper networks is of great interest for future work, and should be possible given recent convex reformulations of deep MLPs [1] under path regularization. For the present paper, we show experimentally that IHT achieves competitive performance for training sparse 3-layer networks following a layerwise optimization strategy. This result is encouraging, though we expect even better performance could be possible by formulating a bespoke sparse IHT for deeper networks (instead of layerwise training). We are excited to pursue this in future work.
>
> [1] Ergen, Tolga, and Mert Pilanci. "Path regularization: A convexity and sparsity inducing regularization for parallel ReLU networks." Advances in Neural Information Processing Systems 36 (2023): 59761-59786.
>
> > Extension to non-Gaussian data:
>
> We focus on Gaussian data to make the analysis tractable, though we agree that MNIST and other images are not Gaussian-distributed. However, in many cases real-world structured, non-Gaussian data (such as image pixel values) is first whitened in a preprocessing step to make the covariates mimic a standard Gaussian distribution, as this is often easier for models to learn with. We hope that future work can extend our analysis to better understand what types of data correlation are benign versus problematic for sparse MLP weight recovery, but we believe that the prevalence of data whitening in practice means that even our current result on Gaussian data is not far from practical relevance.

---

### Official Review · Reviewer_4D7G · 2025-11-06

**Soundness:** 4
**Presentation:** 3
**Contribution:** 4
**Rating:** 8
**Confidence:** 4

**Summary:**

This paper provides the first recovery guarantee for sparse ReLU neural networks by reformulating sparse MLP training as a structured compressed-sensing problem. Under Gaussian data and enumerated activation patterns, the sensing matrix satisfies restricted strong convexity and smoothness, allowing Iterative Hard Thresholding (IHT) to exactly recover sparse weights with high probability. The method requires memory linear in the number of nonzeros. Experiments on planted MLPs, MNIST, and implicit neural representations validate the theory and show that IHT performs comparably or better than Iterative Magnitude Pruning (IMP) while being significantly more memory-efficient.

**Strengths:**

A rigorous and elegant theoretical framework connecting sparse neural network recovery with compressed sensing, supported by solid experimental validation.
1. First work proving exact recovery (Theorem 1) for sparse MLP weights with explicit high-probability bounds and convergence rates; recovery builds on Assumption 2 and the RSC/RSS properties (Lemma 1).
2. Convex reparameterization (Equation 3) via weight fusion turns the nonconvex problem into a linear sensing framework; Assumptions 1–2 are well-motivated and verified through constructive examples in Appendix D.
3. Memory scales linearly with sparsity (s); sample complexity grows with active weights rather than total parameters. IHT achieves competitive or superior accuracy to IMP while using less memory.
4. Well-structured from motivation through theory to experiments, with a comprehensive appendix providing implementation and proofs.

**Weaknesses:**

The main limitation lies in the unquantified theory–practice gap and missing diagnostics connecting empirical behavior to theoretical parameters.
1. Theorem 1 assumes a fixed, enumerated sensing matrix A, but experiments adopt sequential convex updates—updating A after each IHT iteration (following an initial fixed epoch). The paper acknowledges this deviation (Appendix A) but provides no convergence analysis or ablation on update frequency.
2. Empirical values of the RSC and RSS constants and their ratio are not reported, leaving unclear how tight the theoretical conditions are in practice.
3. Results are averages over three runs without error bars, standard deviations, or variance. Adding uncertainty plots (e.g., box plots or success-probability curves) would strengthen performance claims.

**Questions:**

1.Can you analyze how A evolves during sequential convex updates—for example, by measuring similarity between consecutive A matrices—and test how update frequency affects convergence?

2.Can you report empirical ($\alpha,\beta,\beta/\alpha$) for at least one representative setting to show whether the RSC/RSS conditions approximately hold in practice.

---

> ### Author Response · Authors · 2025-11-19
>
> We thank the reviewer for their thoughtful and positive engagement with our work. We are happy to respond to the reviewer’s comments and questions below, and to answer any further questions that may arise during the discussion period.
>
> >Convergence analysis or ablation on sequential convex updates to A, instead of full enumeration of patterns:
>
> Thanks for the suggestion; this analysis was quite interesting! We investigated convergence of A in the following setting: fitting a scalar-output planted sparse 2-layer MLP with input dimension 100, hidden dimension m=10, and sparsity level s=500, optimized over 100 steps of IHT with random initialization and sequential convex updates to A. We update the A matrix every k steps, and report how the PSNR (fitting quality metric, higher is better) changes as a function of k. Note that when k=100, A is never updated because this experiment only runs for 100 steps. We see that when A is updated fairly frequently (k<5) IHT converges and matches the planted model up to numerical precision. However, as A is updated less frequently, the model no longer converges within 100 steps, and PSNR degrades with increasing k.
> | k   | PSNR   |
> |-----|--------|
> | 1   | 161.44 |
> | 2   | 161.44 |
> | 3   | 161.44 |
> | 4   | 161.44 |
> | 5   | 36.63  |
> | 6   | 31.85  |
> | 7   | 31.84  |
> | 8   | 28.49  |
> | 9   | 29.88  |
> | 10  | 28.38  |
> | 15  | 27.23  |
> | 20  | 25.85  |
> | 25  | 24.75  |
> | 30  | 24.74  |
> | 35  | 23.32  |
> | 40  | 23.35  |
> | 50  | 22.02  |
> | 60  | 21.95  |
> | 70  | 21.96  |
> | 80  | 21.97  |
> | 90  | 21.89  |
> | 98  | 21.57  |
> | 100 | 19.88  |
>
> We also evaluate the convergence of A more directly, by checking the first step at which the support of A (which columns are active) stops changing. We find that as k increases, A takes longer to converge. As long as k<5, A still converges within 100 steps. However, when k=5, A does not converge within 100 steps, likely explaining why there is a dramatic drop in PSNR when k increases from 4 to 5. If we instead continue optimization longer when k=5, A does converge at step 135, after which the PSNR increases to 161.44 (corresponding to machine precision). From this set of experiments, we see that the frequency of sequential convex updates to A controls the convergence rate of IHT. This experiment also suggests a natural stopping criterion for IHT: when the support of A stops changing (though in our main experiments we use a fixed number of steps, for fair comparison to IMP).
> | k   | convergence step   |
> |-----|--------|
> | 1   | 43 |
> | 2   | 64 |
> | 3   | 75 |
> | 4   | 84 |
>
>
> >Empirical values of RSC and RSS constants:
>
> It is difficult to compute $\alpha, \beta$ numerically due to the combinatorial nature of the sparsity restriction, but we can give a few examples of our theoretical bounds to build intuition (though these theoretical bounds may be pessimistic, as the reviewer notes). For example, if the error probability $\delta = 0.001$, the neuron separation parameter $\gamma=0.1$, the number of nonzero weights $s=0.05n$, and the dataset size $n=2000$, we have $\alpha \approx 0.027$ and $\beta \approx 95$. As expected, the problem becomes better conditioned ($\beta$ closer to $\alpha$, resulting in faster convergence) if the planted weights are more sparse; if we keep the same settings as above but with $s=0.01n$, $\alpha$ is unchanged but $\beta \approx 19$. We emphasize that our theoretical result ensures convergence regardless of the precise values of these parameters, which only affect the rate of convergence.
>
> >Uncertainty/Variability metrics for experiments:
>
> Thanks for the suggestion to report variability across the three random trials in each experiment. We have plotted heatmaps for each experiment that visualize the standard deviation over the three random trials. Results are here: https://imgur.com/a/1sq5FI9. In some cases we see higher variability near what appear to be phase transitions in the performance for both IHT and IMP, but in other cases results are quite consistent even as quality transitions.

---

### Official Review · Reviewer_RfMf · 2025-11-08

**Soundness:** 3
**Presentation:** 3
**Contribution:** 3
**Rating:** 6
**Confidence:** 4

**Summary:**

This paper provides theoretical guarantees for recovering the sparse weights of a ReLU network using an efficient, memory-linear algorithm. The proposed method, validated on tasks on MNIST, matches or exceeds the performance of a memory-inefficient pruning baseline.

**Strengths:**

1. This work establishes the sparse recovery result for ReLU MLPs.

2. Using shallow networks on Gaussian data, the experiments demonstrate that Iterative Hard Thresholding (IHT) is more effective and memory-efficient than a strong IMP baseline, recovering superior sparse networks.

**Weaknesses:**

Using the eq. (2), training the 2-layer MLP with MSE loss is a nonconvex optimization problem.  From the lines 162-175, the authors consider fixed generator vectors $h_i\in R_d$ and fusing the weights via $w_i=u_iv_i$ to build the eq. (3). This transformation allows the results to be readily obtained using sparse recovery theory. However, several issues arise:

1. The theory requires the matrix $A$ to satisfy the conditions of Lemma 1, which is attributed to the sparsity of $h_i$ or $u_i$. Typically,
 $X$ is assumed to be a Gaussian data matrix. How is Lemma 1 satisfied when the input data is non-Gaussian, such as the binary-valued pixels in the MNIST dataset

2. The theoretical results are promising, but how does the method perform on more complex, real-world datasets like CIFAR-10 or ImageNet-200?

3. This work focuses on recovering the sparsity pattern of the weights $u_i$. What can be said about the sparsity or recovery guarantees for the corresponding weights $v_i$?

**Questions:**

please see weaknesses.

---

> ### Author Response · Authors · 2025-11-19
>
> We thank the reviewer for their thoughtful and positive engagement with our work. We are happy to respond to the reviewer’s comments and questions below, and to answer any further questions that may arise during the discussion period.
>
> > Lemma 1 under non-Gaussian data:
>
> We focus on Gaussian data to make the analysis tractable, though we agree that MNIST and other images are not Gaussian-distributed. However, in many cases real-world structured, non-Gaussian data (such as image pixel values) is first whitened in a preprocessing step to make the covariates mimic a standard Gaussian distribution, as this is often easier for models to learn with. We hope that future work can extend our analysis to better understand what types of data correlation are benign versus problematic for sparse MLP weight recovery, but we believe that the prevalence of data whitening in practice means that even our current result on Gaussian data is not far from practical relevance.
>
> >More complex datasets like CIFAR and ImageNet:
>
> We agree that MNIST is largely a “toy” dataset at this point, and our experiments are intended as empirical validation and extension of our theory rather than as standalone results. The reason our classification experiments focus on MNIST rather than a more complicated image dataset is that our theory is so far restricted to multilayer perceptrons (MLPs), rather than other architectures such as convolutional or attention layers. MLPs can achieve good performance on MNIST (and as implicit neural representations), making these a reasonable test bed for our comparison of different sparse MLP training strategies. For CIFAR and ImageNet classification, convolutional or transformer architectures are usually required for good accuracy, so we defer experiments on those datasets until future work can extend our analysis to those architectures. We agree this is a very exciting direction for future work.
>
> >Recovery of $v_i$ in addition to $u_i$:
>
> Theorem 1 guarantees recovery of the weights $w_i$, where $w_i = u_iv_i$ is a fused version of the first and second layer weights $u_i$ and $v_i$. The sparsity pattern in $w_i$ tells us which $u_i$ in $A$ are active, and then by comparing these active $u_i$ with the corresponding $w_i$ we can extract the scalars $v_i$, which are the unique values such that $w_i = u_i v_i$.

---

### Meta-Review · Area_Chair_Tnzf · 2026-01-08

**Summary:**

Reviewers broadly found the work theoretically rigorous and novel in establishing exact sparse recovery guarantees for two-layer ReLU MLPs via a compressed sensing lens. The main concerns centered on the scope of the theory, restricted to shallow, scalar-output networks and Gaussian inputs, and the theory-practice gap, including sequential updates of the sensing matrix, lack of empirical RSC/RSS diagnostics, and limited baselines. Importantly, these were viewed as limitations of generality rather than flaws in correctness, and the contribution was consistently judged valuable.

**Reviewer Concerns:**

Most reviewers questioned whether the proposed theory can be extended to non-Gaussian or sub-Gaussian settings, or to whitened real-world data. Several also requested experimental validation beyond MNIST, such as on CIFAR-100 or ImageNet-200. These key concerns were not directly addressed in the revision and were instead deferred as potential directions for future work.

**Reviewer Scores:**

Most of the reviewers would have retained or slightly reduced their scores since the rebuttal did not offer results for several practical queries.

---

### Decision · Program_Chairs · 2026-01-26

Accept (Poster)